# Treatment selection using prototyping in latent-space with application to depression treatment

**Akiva Kleinerman**[1]*, **Ariel Rosenfeld**[1], **David Benrimoh**[2,3], **Robert Fratila**[3], **Caitrin Armstrong**[3], **Joseph Mehltretter**[3], **Eliyahu Shneider**[1], **Amit Yaniv-Rosenfeld**[4,5], **Jordan Karp**[6], **Charles F. Reynolds**[7], **Gustavo Turecki**[3], **Adam Kapelner**[8]

**1** Bar-Ilan University, Ramat-Gan, Israel, **2** McGill University, Montreal, Canada, **3** Aifred Health, Montreal, Canada, **4** Shalvata Mental Health Center, Hod Hasharon, Israel, **5** Tel-Aviv University, Tel-Aviv, Israel, **6** University of Arizona, Tucson, Arizona, United States of America, **7** University of Pittsburgh, Pittsburgh, Pennsylvania, United States of America, **8** Queens College, New York City, NY, United States of America

\* akiva.kleinerman@biu.ac.il

**Data Availability Statement:** The synthetic data, described in experiment 1 is public and so is the script used for generating the data: https://github.com/Aifred-Health/DPNN_Experiment. This is

## Abstract

Machine-assisted treatment selection commonly follows one of two paradigms: a fully personalized paradigm which ignores any possible clustering of patients; or a sub-grouping paradigm which ignores personal differences within the identified groups. While both paradigms have shown promising results, each of them suffers from important limitations. In this article, we propose a novel deep learning-based treatment selection approach that is shown to strike a balance between the two paradigms using latent-space prototyping. Our approach is specifically tailored for domains in which effective prototypes and sub-groups of patients are assumed to exist, but groupings relevant to the training objective are not observable in the non-latent space. In an extensive evaluation, using both synthetic and Major Depressive Disorder (MDD) real-world clinical data describing 4754 MDD patients from clinical trials for depression treatment, we show that our approach favorably compares with state-of-the-art approaches. Specifically, the model produced an 8% absolute and 23% relative improvement over random treatment allocation. This is potentially clinically significant, given the large number of patients with MDD. Therefore, the model can bring about a much desired leap forward in the way depression is treated today.

## 1 Introduction

Precision Medicine (often referred to as Personalised Medicine, or PM for short) seeks to customize healthcare, with medical diagnosis, prognosis, treatment and other practices being tailored to the individual patient. This is juxtaposed with the traditional "one-drug-fits-all" model [1]. At the heart of much of PM research and practice stands the challenge of effective personalization, such as selecting an optimal treatment for each individual patient. The basic premise of PM is that patients may vary in their responses to different courses of treatments

mentioned in the article and the rebuttal. The MDD Data used for this study does not belong to the Authors and they are not legally permitted to share it. As such we have updated the data availability statement as follows: "The Data was provided from two sources. The first is the dataset of the IRL-GREY study and those interested in using this data should contact the Institutional Review Board of the University of Pittsburgh at askirb@pitt.edu. The other data source was the NIMH: Data and/or research tools used in the preparation of this manuscript were obtained from the National Institute of Mental Health (NIMH) Data Archive (NDA). NDA is a collaborative informatics system created by the National Institutes of Health to provide a national resource to support and accelerate research in mental health. Dataset identifier:10.15154/1523049. This manuscript reflects the views of the authors and may not reflect the opinions or views of the NIH or of the Submitters submitting original data to NDA. Those wishing to use this data can make a request to the NIMH (visit https://nda.nih.gov/)".

**Funding:** AK and AR were supported by the Chief Scientist Office, Israeli Ministry of Health (CSO-MOH, IL url: https://www.health.gov.il/English/Pages/HomePage.aspx) as part of grant #3-000015730 within Era-PerMed. DB, CA, JM, RF and GT were also funded by the Canadian arm of this grant (ERA-Permed Vision 2020 supporting IMADAPT), with DB, CA, JM, RF funded via their involvement in Aifred Health, which was subcontracted to complete work as part of this grant. This was the grant which served as the primary funder of this work. The funders (the granting agencies) had no role in study design, data collection and analysis, decision to publish, or preparation of the manuscript.

**Competing interests:** AK, AK and AR have received honoraria from Aifred Health (https://www.aifredhealth.com/). Aifred Health was not the primary funder of this study, and honoraria were provided in connection to the support of the Aifred Health team during the IBM Watson AI XPRIZE competition. Aifred Health-affiliated co-authors collaborated with other co-authors in the conduct of this work. JK is a member of Aifred Health's scientific advisory board and has received stock options from Aifred Health. DB, CA, RF, JM are shareholders, employees and/or officers of Aifred Health. AK, AK, AR, CA, JM, RF, and DB are co-inventors on a patent pending relating to this work." This does not alter our adherence to PLOS ONE policies on sharing data and materials. Data is not owned by the authors, but information on how to request it is available via the data sharing

and, as such, may benefit more from different courses of treatment. Recently, this has been increasingly achieved with the use of machine learning models.

Within the PM realm, machine-assisted treatment selection approaches can be largely classified into one of two paradigms. The first is the *Fully personalized* paradigm, where historical data is used to identify the relationships between patients' characteristics, treatments, and medical outcomes in order to determine the optimal course of treatment of new patients. Using this paradigm, a model is commonly trained to approximate the individual treatment effects of the available treatments and an optimal treatment for each patient is derived from this model [2, 3]; The second is the *Sub-grouping* paradigm, where groups of "similar" patients are identified and then associated with an optimal treatment at the group level. Using this paradigm, some clustering algorithm is used in order to identify the clusters and then each cluster is associated with an optimal treatment [4, 5]. Sub-grouping is often referred to as *prototyping* since each group of patients is commonly represented by one or more (real or virtual) "prototypical patients" which are in turn associated with an optimal treatment.

Algorithms from both treatment selection paradigms have demonstrated significant benefits across a wide range of medical applications. However, in some medical settings, both paradigms suffer from important limitations: For example, the fully personalized paradigm is commonly focused on minimizing the prediction error over the individual treatment effects [2, 4]. However, in practice, one is commonly more interested in identifying the "best" course (s) of treatment. These two objectives do not necessarily coincide when the probability estimations are inherently inaccurate. Specifically, we are generally more interested in the induced *ranking* over the possible treatments rather than the accuracy of the probability estimations themselves, as the objective of treatment selection is to choose the best possible treatment from a given set (i.e., the set of possible treatments). This concern is mitigated in some domains by adopting a sub-grouping paradigm (e.g., [4]). Introducing subgroups makes it easier to identify which patient groups may benefit the most from different treatments. However, the sub-group paradigm introduces new shortcomings: first, defining "good" sub-grouping criteria is challenging, especially when the pathophysiology of the disease and their relation to treatment outcomes are unknown, as is the case in many psychiatric disorders [6]. Second, group-level treatment selection naturally translates into some loss in accuracy when the groups are not entirely cohesive (e.g., [7]).

In this article, we propose a novel deep learning-based approach that strikes a balance between the two treatment selection paradigms by *simultaneously* identifying prototypes (subgroups) of patients as well as approximating outcome prediction in a personalized manner. More specifically, our approach aims at finding "actionable" prototypes, meaning that they differ not only in their characteristics but, importantly, in their expected responses to the available courses of treatment. Our approach uses a novel deep-learning architecture and a multifaceted loss function which balances between the accuracy of the prediction on the individual level and the cohesiveness of the identified prototypes in terms of predicted treatment outcomes. Importantly, the model outputs not only group membership for a given patient, but also personalized probabilities of treatment success for each possible treatment.

Our approach performs the prototyping identification in the latent space and, as such, it is specifically suited for domains where a useful sub-grouping of patients is assumed to exist, but is not observable in the non-latent space. Note that in some medical domains this assumption may prove irrelevant as patients can be divided into cohesive sub-groups in terms of treatment responses by observed features alone. For example, the optimal course of treatment for leukemia (and several other forms of cancer) is commonly determined by age group and other known factors [8]. Nonetheless, in several other medical domains such as psychiatry, such sub-grouping in the non-latent space is found to be only weakly connected with treatment

statement. Sufficient information is available in the manuscript and in the associated appendices and code to reproduce the experiments described herein.

response (see [9, 10] for a discussion). As such, sub-grouping in observable space may not be optimal in mental health.

## 1.1 Treating major depressive disorder

For evaluating our proposed approach, we focus on the personalization of treatment for Major Depressive Disorder (MDD). MDD is highly prevalent in the general population and is associated with grave consequences, including excessive mortality, disability and secondary morbidity. According to the World Health Organization (WHO), more than 300 million people were affected by depression worldwide in 2017 [11]. In the United States, MDD was a leading cause of disability in all ages in year 2018, substantially more than most other physical and mental conditions [12]. Consequently, the economic burden of MDD on society is very high [12, 13]. MDD is diagnosed based on a heterogeneous group of symptoms, and two patients with depression can have very different clinical phenomenology. The DSM-5 criteria for depression include loss of interest or pleasure in usual activities, depressed mood, increased or decreased appetite or weight, increased or decreased psychomotor activity, increased or decreased sleep, fatigue, poor concentration, feelings of guilt or worthlessness, and suicidality; a patient must have at least five of these symptoms and at least one of these must be depressed mood or reduced interest or pleasure [14]. The precise pathophysiology of MDD has yet to be fully elucidated [15].

Antidepressants are the most common treatment for MDD and are among the most prescribed medications [16]. While they have demonstrated effectiveness, MDD patients vary significantly in their response to the various treatments. The current status quo for MDD treatment is an educated trial-and-error approach in which patients typically undergo several rounds of different antidepressants [17]. In clinical practice, a single treatment course typically lasts six months or more and roughly one third of patients do not respond to treatment following an adequate trial. This means that they can be forced to go through several unsuccessful rounds of drug treatments, with this effort at times extending over several years [18]. This process has severe psychological, economic and social consequences for both patients and their families [19, 20]. It is commonly assumed that MDD patients can be categorized into useful sub-groups on the basis of presenting symptoms that may be associated with differential treatment response (e.g., [4, 21–23]). Unfortunately, commonly used clinical sub-groups of depression that are defined in the non-latent space (i.e., atypical, melancholic or anxious prototypes) have failed to predict any significant difference among currently available antidepressant medications [9].

This need for improved treatment selection in depression, combined with the possibility that useful sub-groups for treatment selection may exist in the latent space, make MDD an excellent candidate disorder to test our approach. To do so, we will use both synthetic data and clinical data. Our clinical dataset combines data from several clinical trials, and describes 4754 MDD patients who were treated as part of clinical trials of antidepressant treatment. Each patient in these datasets is described by sociodemographic information and clinical symptomatology at baseline, the treatment they received, and the outcome of the treatment after 12 weeks. Below, we describe the datasets in further detail (Subsection 5.4.1 and Appendix B). The synthetic data, described below, has similar characteristics to the clinical dataset. Using both of these datasets, we demonstrate our approach's added value compared to state-of-the-art methods aimed at improving treatment selection for MDD. Given that the current standard of treatment for many psychiatric disorders, including MDD, is an educated "trial-and-error" approach [24], our approach could help bring about a much desired leap forward in treatment effectiveness in terms of increased remission rates and reduced length of the process of finding

optimal treatments at the individual patient level. In turn, this can translate to a reduced social and economical burden from MDD in the population level.

## 2 Related work in machine-assisted treatment selection

Automated treatment selection is an evolving field which has recently garnered increased interest from researchers [2] and by the media [25]. As discussed above, AI-based treatment selection in healthcare is commonly addressed through two different computational paradigms: the fully personalized and sub-grouping approaches. Both approaches have had success. For example, for eradicating brain tumors, a fully-personalized approach has been shown to effectively maximize the expected clinical benefits while minimizing side effects [26]. With respect to subgrouping approaches, in [27], the authors suggested clusters based on non-adherence for psychosis treatment. More recently, researchers have shown that these clusters demonstrate significant differences in terms of re-hospitalisation rates and maintenance of the original medication [28]. However, as described above, both approaches have significant limitations. Fully personalized approaches can focus too much on maximizing accuracy when the desired output is differential treatment benefit and they do not utilize potentially useful information related to patient subgroups; and, by design, the sub-grouping paradigm does not explicitly capture the possibly complex links between individual patients' features, treatment and possible outcomes, and it does not provide explicit predictions for expected outcome on an individual level. In addition, in many cases, the sub-groups can overlap such that a single patient may be associated with more than one cluster. Therefore, a simple mapping from clusters to treatments may not suffice.

In order to mitigate these limitations we propose a novel model, which we will call Differential Prototypes Neural Network (DPNN for short). DPNN is proposed as a middle-ground solution between the two existing paradigms discussed above. Similar to the fully-personalized paradigm, during training the DPNN considers the outcome prediction accuracy on an individual level. At the same time, it takes inspiration from the sub-grouping paradigm by performing prototyping which directly feeds into the individual level prediction. Unique to our approach is simultaneous optimization of both the sub-groupings and the individual predictions.

Unlike traditional unsupervised clustering methods, the prototypes are constantly tuned during the training process in order to guarantee that the prototypes will approximate an optimal treatment selection policy. To the best of our knowledge, this is the first work that applies such an approach in the field of PM. We will now discuss similar approaches in previous studies.

A Similar approache to DPNN is the work described by Ross et al. [7], that addresses effective pediatric asthma treatment. This approach also combines both the fully personalized and sub-grouping paradigms. However, unlike DPNN, which simultaneously considers sub-grouping and personalized outcome prediction accuracy, Ross et al. perform these tasks sequentially. Specifically, their approach divides the patients into clusters and later assigns different outcome prediction models for each possible sub-group. The process is then continued until there are no prediction performance improvements that can lead to further changes in clusters. Another difference between the approaches is that DPNN is also suited for treating patients who can be associated with more than one cluster, as is the case in many medical settings including depression treatment [9].

Our approach draws inspiration from the models proposed in [29, 30] for promoting interpretability in classification tasks. In these works, neural network architectures are utilized for partitioning samples into sub-groups in various classification tasks. Unlike in our

approach, these prototypes have not been linked, explicitly or implicitly, to optimal interventions or treatments. Our task, is to improve decision-making, which requires the linkage of subgroups to outcomes (given treatments). As such, our task leads to fundamental differences in network architecture from previous work and the development of a suitable loss function as discussed in Section 4.1.

In order to better situate our proposed approach in the field and evaluate its potential benefits over the standard "fully personalized" and "sub-grouping" paradigms, we adopt a few key representatives of each approach for comparison and discuss them next. These representatives are also used in our experimental evaluation in Section 5.

Starting with the fully personalized paradigm, we use three representatives: 1) A state-of-the-art method for estimating Individualized Treatment Effects (ITE) called CFRnet [31]; 2) A state-of-the-art deep-learning model explicitly designed for depression treatment selection called Vulcan [32]; and 3) A classic Case Based Reasoning (CBR) approach [33]. We discuss these representatives next and contrast them with our approach.

Methods for estimating ITE, and specifically CFRNet, focus on leveraging available clinical data for predicting outcomes of treatments [34, 35]. This line of research mainly addresses the lack of counterfactual data in clinical results, meaning that the available data consists of only the outcome of a single administered treatment for each patient. Recent advances in this line of work have investigated methods that focus on balancing the distributions of control and treated groups (e.g., [36]). It is important to note that our task is slightly different for two main reasons: 1) We assume minimal selection bias in the data, given that the data was collected from clinical trials in which the patients were assigned randomly to the different treatment conditions or were all assigned to the same treatment for each patient; and 2) Our main objective is not (solely) to accurately predict or quantify the effects of possible treatments, but rather to find the best treatment(s). In other words, the correct ranking of the treatments is the main objective, as opposed to solely maximizing prediction accuracy. The CFRnet method [31], is a prime example of this approach which uses a deep learning framework for counterfactual regression, that simultaneously fits a balanced representation of the data and an hypothesis for the outcome of counterfactuals.

Vulcan is a deep-learning based treatment selection component of the Aifred system [37] and is considered to be the state-of-the-art representative of the fully personalized paradigm for MDD treatment selection. Vulcan is explicitly designed for depression treatment selection and was recently evaluated on a subset of the MDD data set used in this study as well (see Section 5.4.1). This evaluation revealed that it is superior to other machine learning based approaches such as random forests [32]. Similar to the CFRNet method, Vulcan uses a neural network based model that is fed with the features of a patient and a specific treatment and outputs the remission probability associated with that treatment.

Case-based recommender systems (CBR) is a private case of the Recommender Systems [38] approach which seeks to select the best item(s) from a set of possible items for a given user, according to his or her estimated preferences. In analogy to our task, a patient can be viewed as a "user", the treatments can be viewed as "items" and the "preferences" in our context would be the suitability of the treatment to the patient. However, most common approaches for RS (e.g. collaborative filtering, content based) require many interactions between users and items [39], while in clinical data each patient is often administered only a single treatment and experiences only a single outcome. Nevertheless, several researchers have investigated RS that utilize the CBR approach for treatment selection (e.g. [33]) which relies on the idea of detecting similarities between patients. These have also been applied for mental health treatment selection [40].

For the subgrouping paradigm, we use two representatives: 1) A standard K-means clustering algorithm [41]; and 2) A state-of-the-art depression sub-grouping approach based on latent profile analysis [4]. We discuss these representatives next and contrast them with our approach.

The most straight-forward method to perform sub-grouping is to implement one of the classic clustering algorithms available today [42]. In this work, we chose the popular K-Means algorithm in order to cluster the patients. We represented each patient with the relation to the K-Means clusters' centroids and fed each patient to a separately trained fully-connected classification neural network in order to predict the outcome of the treatment. We term this method as KMNN.

A more advanced implementation of the subgrouping paradigm is introduced by Saunders et al. [4, 23], who have shown that using Latent Profile Analysis (LPA) [43], MDD patients can be divided into eight sub-groups that differ significantly in their reaction to treatments and specifically in the most effective treatment. This method, which we will refer to as LPAD (Latent Profile Analysis for Depression treatment), is considered the state-of-the-art representative of the sub-grouping paradigm.

## 3 Problem definition

Let us define our treatment selection problem setting more formally.

We are given a data set of $N$ samples $D = \{(x_1, t_1, y_1) \ldots, (x_N, t_N, y_N)\}$, where $x_i$ describes a patient sampled from a given distribution $\chi$ and represented as a $d$ dimensional feature vector $x_i \in R^d$, $t_i \in T$ indicates the treatment received by patient $x_i$ from a finite set of $k > 1$ treatment options, and $y_i \in Y$ indicates the observed outcome of the treatment. Importantly, we assume $D$ resulted from an *unbiased treatment selection*, such as a controlled clinical trail. Namely, the assignment of each patient $x_i$ to $t_i$ was independent of $x_i$. Formally, for any $t \in T$, $p(t|x_i) = p(t)$.

We assume that $Y$ is *binary* and consists of only a desired outcome and a non-desired outcome. Adopting the terminology of our application domain (i.e., MDD treatment), we refer to the desired outcome as *remission*, denoted as $r$, and a non-desired outcome as non-remission, denoted $\bar{r}$. It is important to note that our formulation and approach can be readily adapted to any outcome space of choice and the above restriction is for presentation and evaluation purposes only.

Assuming there are $k$ potential treatments, for a given patient $x_i$, there are $k$ corresponding *potential* binary outcomes: $y_i^{(0)}, y_i^{(1)}, \ldots, y_i^{(k-1)}$, where $y_i^{(j)} \in \{0, 1\}$. However, our data consists of only a single *observed outcome* for $x_i$ who received $t_i$, namely $y_i^{(t_i)}$. In other words, $D$ does not include the counterfactuals—namely, the outcomes of non-received treatments $t \neq t_i$.

The optimal treatment selection policy, $\pi^*$, assigns $t_i^*$ for each patient $x_i$ such that it maximizes the desired outcome probability (i.e., remission). Formally, for a patient $x$,

$$\pi^*(x) = \underset{t \in T}{\operatorname{argmax}} \ Pr(r|x, t) \tag{1}$$

where $Pr(r|x, t)$ is the probability of remission for patient $x$ given treatment $t$. Naturally, the true probability is unknown.

## 4 Approach

Since $Pr(r|x, t)$ is unknown, deriving an optimal treatment selection policy as defined in Section 3 is very complex. Similar to other treatment selection techniques proposed in the literature (see Section 2), we tackle this challenge by approximating $Pr(r|x, t)$ such that $\widehat{Pr}(r|x, t) \sim Pr(r|x, t)$. However, our true objective is to "approximate" the optimal policy $\pi^*$

by deriving a treatment selection policy as follows:

$$\pi(x) = \underset{t \in T}{\mathrm{argmax}} \; \widehat{Pr}(r|x, t) \tag{2}$$

By approximating the optimal policy we mean that we seek to minimize the following loss function:

$$Loss(\pi) = \underset{x \sim \chi}{\mathbb{E}} [Pr(r|x, \pi(x)) - Pr(r|x, \pi^*(x))] \tag{3}$$

However $Pr(r|x, t)$ is unknown. As such, any approximation thereof need not necessarily minimize the above loss. To overcome this hurdle, we propose to approximate $Pr(r|x, t)$ in an *unorthodox way* such that it would potentially prove more useful for minimizing the above loss *indirectly* compared to standard approximations to $Pr(r|x, t)$. Our method leverages the assumption that patients may be divided into sub-groups which vary significantly in their reactions to treatments as discussed in Section 2. In this work, we implement our approximation approach with a neural network based architecture, that: 1) identifies prototypes of patients; and 2) predicts the remission probability for each patient-treatment pair based on the their resemblances to identified prototypes. We discuss our proposed neural architecture and loss functions in following subsections.

### 4.1 Model architecture

Our proposed neural network architecture consists of the following three main components (which will be explained thereafter):

1. A symmetrical autoencoder, including an encoder—$e$: $R^m \to R^q$, and a decoder—$d$: $R^q \to R^m$.

2. A prototype layer, $g$.

3. A classification network $h$: $R^q \to R^K$.

The architecture uses an autoencoder in order to produce features in the latent-space, $R^q$. We denote the dimension of the original data as $m$ and the dimension of encoded data as $q$. The encoded data is used for both finding prototypes and to calculate the outcome prediction for each patient separately. Specifically, for each patient $x_i$, the network first encodes the $m$ features of $x_i$ into an encoded representation denoted as $e(x_i)$. $e(x_i)$ is used in two ways: 1) It is used together with the decoder $d$ in order to complete the auto-encoding process and; 2) It is fed into the prototype layer.

The prototype layer $g$ consists of $\ell$ randomly initiated prototypes $P = \{p_1, p_2, \ldots p_\ell\}$ each of $q$ dimensions. In this layer, the architecture calculates the distance between the encoded sample $e(x_i)$ and each of the prototype vectors using some distance measure. These distances, denoted as $d_i = \{d_i^1, d_i^2 \ldots, d_i^\ell\}$, have practically reduced each patient's representation to an $\ell$-dimensional encoding. Finally, the resulting encoding is fed together with the treatment representation ($t \in T$) into a standard classification network, $h$.

The classification network $h$ is a standard fully connected neural network, used for obtaining a probability distribution over the possible outcomes. During the training process, after obtaining these probabilities, the network calculates the loss, as described below (Section 4.2), and then back-propagates the loss value in order to tune all the components of the network, including the prototypes. By doing so, the prototypes are tuned in order capture differences in the treatment effect. A visualization of the entire architecture is given in Fig 1.

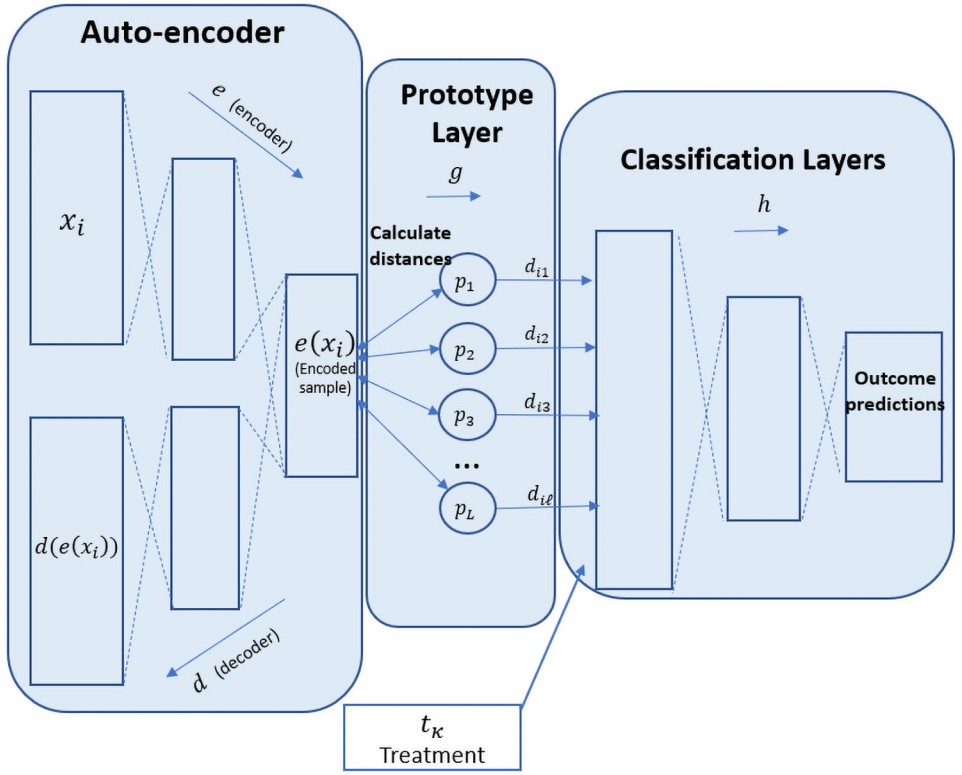

**Fig 1. Visualisation of the DPNN architecture.**

Note that the prototype vectors are randomly initialized and their values are tuned during the training process with an appropriate loss functions which will be discussed next. The number of prototypes, $\ell$, is predefined prior to the training process similar to other standard clustering-based approaches [41].

## 4.2 Objectives and loss functions

Recall that our main objective is to determine the most appropriate treatment(s) for each patient. Our proposed approach seeks to accomplish that by approximating the probability distribution of each patient-treatment pair in a way which will *indirectly* pursue this main goal. As discussed before, we propose to accomplish this by leveraging the assumption that patients may be divided into "actionable" subgroups which differ in their responses to different treatments.

With the above motivation in mind, in addition to the standard accuracy objective, we incorporate an additional objective which is to increase the discordance between the prototypes in respect to their expected outcomes across the possible treatments. Thus, our approach could potentially reveal prototypes that will prove more useful for treatment selection.

These two objectives are optimized using a complex loss function that consists of the following three components:

1. $L_1$: The accuracy loss evaluates the binary cross-entropy loss in predicting the outcome for the training set. Formally, for a training set $D$,

$$L_1(h \cdot g \cdot e, D) = \sum_{i=1}^{|D|} \mathbb{1}_{[y_i=r]} log(h \cdot g \cdot e(x_i)) + \mathbb{1}_{[y_i=\bar{r}]} log(1 - h \cdot g \cdot e(x_i)) \tag{4}$$

where $h \cdot g \cdot e(x_i)$ is the output of the network for $x_i$ and $\mathbb{1}_{[y_i=r]}$ $(\mathbb{1}_{[y_i=\bar{r}]})$ is an indicator function for the event that the outcome of sample $x_i$ is $r$ $(\bar{r})$.

2. $L_2$: The auto-encoder loss which measures the Euclidean distance between the original samples and the decoded samples as obtained through $e$ and $d$. $L_2$ is defined as

$$L_2(d \cdot e, D) = \sum_{i=1}^{|D|} \| \mathbf{x_i} - \mathbf{d} \cdot \mathbf{e(x_i)} \| \tag{5}$$

By minimizing this loss the model ensures that the encoded data contains meaningful latent features, meaning features that are rich enough to reconstruct the encoded sample as close as possible to the original.

3. $L_3$: Prototype variance loss. This novel component measures the variance in the expected treatments outcomes across all prototypes. $L_3$ is calculated as follows: Each of the $\ell$ encoded prototypes is matched with each of the $k$ possible treatments, and fed through the classification network. As a result, our network produces a matrix, denoted $Y_P$, of size $\ell \times k$, where each cell contains the prediction for one possible combination of prototype and treatment. We will denote the remission probability of prototype $i$ with treatment $\kappa$ as $y_i^\kappa$. $L_3$ is defined as follows:

$$L_3 = -((\alpha)intra_{var}(Y_P) + (1 - \alpha)inter_{var}(Y_P)) \tag{6}$$

where $intra_{var}$ quantifies the variance in predictions *within* the prototypes (across the possible treatments), $inter_{var}$ quantifies the variance in prediction *between* the different prototypes, and $\alpha \in (0, 1)$ is a hyper-parameter that balances between the two types of variance. Formally,

$$intra_{var}\left( \begin{bmatrix} y_1^1 & \cdots & y_1^k \\ & \cdots & \\ y_\ell^1 & \cdots & y_\ell^k \end{bmatrix} \right) = \sum_{l=1}^{\ell} \frac{\sum\limits_{\kappa \in T}(y_1^\kappa - \mu_\ell)^2}{l} \tag{7}$$

where $\mu_\ell$ is the mean probability of remission for prototype $l$ across all treatments. Similarly,

$$inter_{var}\left( \begin{bmatrix} y_1^1 & \cdots & y_1^k \\ & \cdots & \\ y_\ell^1 & \cdots & y_\ell^k \end{bmatrix} \right) = \sum_{\kappa \in T} \frac{\sum\limits_{l=1}^{\ell}(y_l^\kappa - \mu_\kappa)^2}{\ell} \tag{8}$$

where $\mu_\kappa$ is the mean prediction of remission for treatment $k$ across all prototypes.
Notice that the summation of the variance terms is negated in $L_3$ since we are interested in *increasing* the variance combination.
Below we present the motivation for this type of loss and illustrate the difference between both types of variance.

**Table 1. High variance across prototypes.**

| Treatment / Prototype | $T_1$ | $T_2$ | $T_3$ |
|---|---|---|---|
| $P_1$ | 0.3 | 0.25 | 0.7 |
| $P_2$ | 0.35 | 0.2 | 0.65 |
| $P_3$ | 0.8 | 0.02 | 0.03 |

In order to motivate and clearly illustrate $L_3$, we provide two examples. Say our network is configured to find three prototypes ($\ell = 3$), and the dataset includes three possible treatments ($k = 3$). In order to calculate $L_3$, our network will first produce 9 different remission probabilities: 3 for each of the 3 prototypes. Now, say that the network produced the probabilities presented in Table 1.

We can observe that prototypes $P_1$ and $P_2$ are expected to react similarly to the possible treatments, while they both differ significantly from $P_3$. As our goal is to find meaningful prototypes with respect to the treatment outcome, we would like to improve the selection of prototypes by increasing the difference between $P_1$ and $P_2$ while keeping the clear difference between them and $P_3$. We obtain this goal by increasing the variance across the prototype dimension: for each treatment, we measure the variance across all prototypes and calculate the summation of all variances, as defined in Eq 8.

Now say the probabilities for remission were as given in Table 2.

In this example, the prototypes are clearly differentiated with respect to remission rates, but the differences within each prototype across different treatments are minimal. These prototypes are associated with *overall* chances of remission. As such, they are not particularly helpful for improving treatment selection. Therefore, we would like to make sure that the differences between the treatments expected outcomes, within each prototype, would become more significant. Therefore, we would like to increase the variance in the expected outcomes for treatment within each prototype. We denote this type variance as $intra_{var}$ as defined in Eq 7.

As can be seen from the above two examples, both types of variance are essential, yet they are *partially conflicting*. The more the variance across the prototypes is increased, the more chances are that the differences within prototypes decrease, since high variance between prototypes naturally leaves less space for variance within the prototypes and vice-versa. Therefore, these variance components are combined through a weighted summation that allows for a customised configuration between these loss components. This controls the prototypes' training with the objective being that they are sufficiently spread out across the latent space so as to potentially capture the nuances and characteristics of the patient population. A visualization of the computation of $L_3$ is presented in Fig 2.

We linearly combine the three loss components as:

$$L = L_1 + \lambda_1 L_2 + \lambda_2 L_3 \tag{9}$$

where $\lambda_1$ and $\lambda_2$ are hyper-parameters that balance between the different objectives of the network.

**Table 2. Low variance within prototypes.**

| Treatment / Prototype | $T_1$ | $T_2$ | $T_3$ |
|---|---|---|---|
| $P_1$ | 0.81 | 0.8 | 0.83 |
| $P_2$ | 0.45 | 0.44 | 0.47 |
| $P_3$ | 0.19 | 0.2 | 0.18 |

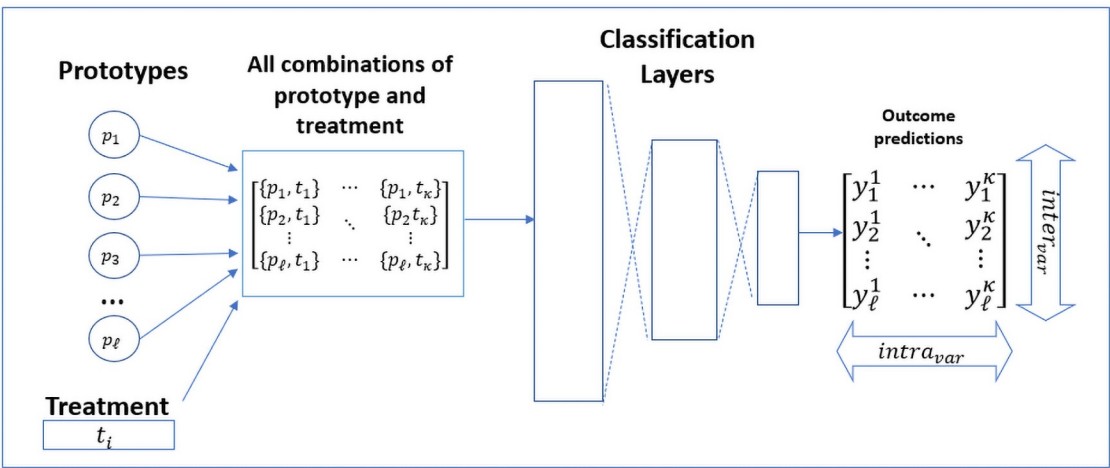

**Fig 2. Visualisation of $L_3$'s computation.**

All of the components described above (the autoencoder, prototypes and classification layers) are trained simultaneously, meaning that after every single feed-forward, the combined loss is back propagated and tunes all the components accordingly.

## 5 Evaluation

In order to evaluate our approach, we performed two experiments: First, we used synthetic data which was created to follow our two central assumptions: 1) Patients can be divided into sub-groups that differ in their reactions to different treatments; and 2) These sub-groups are defined in the latent space. This experiment, which we will refer to as *Experiment 1*, is used for demonstrating the potential benefits of our approach when our assumptions are fully met and counterfactual data exists for evaluation purposes. Second, we use an extensive secondary real-world clinical dataset for MDD treatment selection that combines three major clinical trials ($N = 4, 754$). In this experiment, which we will refer to as *Experiment 2*, we demonstrate the potential benefits of our approach in a real-world, high stakes medical domain in which no counterfactual data exists and our assumptions cannot be definitively confirmed.

In order to properly compare the proposed DPNN approach to the five benchmark approaches discussed in Section 2, several technical amendments were needed. Next, we discuss this slight modifications followed by the results of the two experiments described above.

### 5.1 Benchmarks

We compared our DPNN approach to five representative methods discussed in Section 2:

1. **CFRnet** [31]: a neural network-based model that builds a balanced representation of the data, in order to overcome possible bias between treatment groups, and predicts the outcome of possible counterfactuals. CFRnet was originally designed for data including two groups of treatments and therefore we adjusted the original framework to fit multiple treatment groups, as we describe in Appendix E. CFRnet was recently evalauted in other similar healthcare domains [31, 44].

2. **Vulcan**: a state-of-the-art treatment selection method specifically tailored for MDD [32]. The original code used in [37] was provided by the authors, some of which co-author this article as well. The code was used "as-is" without further modification.

3. **CBR**: the prediction of each treatment *t* is estimated by the *k* nearest neighbors (KNN) in the dataset who received treatment *t*. The similarity distance is based on popular cosine similarity measure and the prediction is calculated as the weighted sum of the neighbors' outcomes weighted by distance.

4. **KMNN**: The algorithm works in two phases, first a standard K-Means algorithm is executed to identify clusters of patients based on their observable features, regardless of treatments and outcomes. Then, for a given patient, distances are calculated with respect to the centroids of the identified clusters which are then fed to the separately trained fully-connected classification network in order to predict the outcome of the treatment.

5. **LPAD**: the state-of-the-art method for MDD treatment selection using sub-grouping [4]. As in DPNN, the clustering is performed in the latent-space. Note that, unlike DPNN, LPAD does not explicitly predict the remission probability but rather focuses on determining optimal treatment. Therefore, a few technical steps were taken in order to appropriately evaluate the LPAD also in terms of accuracy (see Appendix D).

For replication purposes, the Vulcan implementation is publicly available at: https://github.com/Aifred-Health/Vulcan. All other benchmarks, as well as our DPNN implementation and data analysis scripts, are publicly available at: https://github.com/Aifred-Health/DPNN_Experiment.

## 5.2 Evaluation metrics

In many prediction tasks, the *Area Under The Curve (AUC)* is the central metric of choice for evaluating a model's performance. However, for our task, the AUC metric does not fully reflect the quality of the output, since it is *not* a top-biased measure [45]. More specifically, in our setting, our prime objective is to select an optimal treatment which is most likely to provide a desirable outcome. Therefore, errors at the top of the list of treatments, sorted by their likelihood for desired outcome (i.e., either ranking a "bad" treatment at the top of the list or ranking a "good" treatment at the bottom of the list), are more critical than errors in the rest of the list. Through this perspective, our evaluation is akin to that of *learning to rank* tasks, which require different metrics than other prediction tasks do. Therefore, in addition to the AUC metric, we adopt the standard *Mean Reciprocal Rank (MRR)* [46] commonly used in learning-to-rank tasks.

Formally, let $\vec{r_i}$ be the vector of *real* remission probabilities for patients *i* across all possible treatments, and $\vec{pr_i}$ be the vector of *predicted* remission probabilities. MRR is calculated as follows:

$$MRR(S) = \frac{1}{|S|} \sum_{i=0}^{|S|} rank_{\vec{p_i}} \left( argmax(\vec{pr_i}) \right) \tag{10}$$

where *S* is the evaluation set and $argmax(\vec{pr_i})$ is the treatment with the highest remission probability for patient *i*, according to the model's output. $rank_{\vec{p_i}} \left( argmax(\vec{pr_i}) \right)$ is therefore the real rank of selected treatment.

Similarly, we define the following Remission Prediction Loss (RPL) which measures the difference between the true *remission probability* of the selected treatment and the highest *remission probability* across all treatments as follows:

$$RPL(S) = \frac{1}{|S|} \sum_{i=0}^{|S|} \left( max(\vec{r_i}) - r_i^{argmax(\vec{pr_i})} \right) \tag{11}$$

where $r_i^{argmax(\vec{pr_i})}$ is the *true* remission probability of the treatment with the highest *predicted* remission probability.

The MRR and RPL metrics capture different notions for measuring the discordance between the treatment selected and the optimal one. MRR captures this notion in terms of ranking order while RPL captures the same by effectiveness differences. Notice that for the *MRR* metric—the higher the score—the better, while for the *RPL* metric—the lower the score—the better.

Unfortunately, calculating the MRR and RPL can only be done in settings where some counterfactual knowledge is available. For example, MRR can only be calculated if the data contains the true ordering over the possible treatments for patients. While such data is available in Experiment 1 (synthetic data), it is unavailable in Experiment 2 (real-world clinical data). Therefore, in Experiment 2, we adopt a different metric as proposed in other similar medical treatment selection works (e.g., [32]) called *Remission Rate* or RR for short. The idea behind the RR measure is to use the test set such that for each patient in it, the model is executed and the predicted optimal treatment is identified. Then, we filter out all patients in the test set who did not receive the predicted optimal treatment. Through this process, we are left with a subset of the original set of patients, all of whom had received the predicted optimal treatment in practice. Based on these patients alone, the average remission rate, i.e., the portion of remission-labeled patient-treatment pairs, is calculated and reported as the RR of the model. Since we are using a k-fold evaluation technique, we derive a set of RR results, one for each fold. Note that this procedure is not optimal since the RR metric is only based on a (possibly small) subset of patients. In addition, unlike the *MRR* and *RPL* metrics, the RR metric does not reflect the quality of the ranking, which is desirable in our setting. Despite the fact that the RR is not an optimal metric, it is a meaningful one- it represents the proportion of patients who would reach remission if they were assigned treatments using the model, which can then be compared to the baseline remission rate in order to determine the clincal value of the model.

## 5.3 Experiment 1—Synthetic data

**5.3.1 Data generation.**   In this study, we consider settings where patients can be assigned one of multiple possible treatments and: 1) The patients can be divided into sub-groups that differ in their reactions to different treatments; and 2) These sub-groups are defined in the latent space, which is not directly observable by the model, meaning that the features that define these clusters are not given as input to the model. As such, in order to generate appropriate synthetic data that adheres to these assumptions we perform a "reverse engineering"-like process, starting with a set of randomly generated "prototypes", around which fictitious patients are created. We assume four possible treatment exist, and each treatment is associated with a unique function that maps a patient to an outcome. These functions were non- linear and generated randomly. Each patient is randomly assigned to one of four treatments and the associated likelihood of remission. The resulting dataset consists of 10,000 patients which are represented in the slightly noisy *feature space* yet are clustered around 5 prototypes *in the latent space*. The parameters used for our data generation procedure were not meaningful in the context of our task and when those were varied little difference in patterns was encountered. We describe the process in more detail in Appendix C, and the script for generating the data is available in https://github.com/Aifred-Health/DPNN_Experiment.

**5.3.2 Training.**   We separated the dataset into a training set and a test set using the k-fold cross validation technique [47], in which the dataset is split into $k$ consecutive folds and each fold is used once for the test set while the $k-1$ remaining folds are used for the training set.

We report the results for $k = 5$, while similar results were obtained for other standard choices for $k$. For each of the splits, we trained and tested the DPNN and each of the benchmark algorithms in the same way. We repeated this process 100 times in order to obtain a sufficiently large results pool and subsequently obtained 100 samples of each metric for each method.

All clustering algorithms (DPNN, LPAD and Kmean) were executed assuming 5 clusters. We obtained the hyper-parameters by an automated process that searched various combinations of parameters and found the combination that yielded the best results. We ran all models for 100 epochs, with 10 samples in each batch and a 0.0001 learning rate. We found that our network performed best with a single hidden layer in both the autoencoder and the classification network. Appendix A provides the fully hyper-parameter setup we used and the hyper-parameter tuning process.

**5.3.3 Results.** For all examined metrics, the results do not distribute normally according to the Shapiro-Wilk test [48], a standard issue when analyzing the performance of machine learning algorithms [49]. As such, we use the Friedman's test [50] followed by post-hoc Wilcoxon signed rank significance tests (Wilcoxon's test for short) [51] for pairwise comparisons with proper $p$-value adjustment using Bonferroni correction [52].

The Friedman's test showed that the evaluated methods differ significantly on all examined metrics (MRR, RPL and AUC), $p \leq 0.01$.

For the MRR metric, we found the DPNN model was significantly superior to all the benchmarks ($p \leq 0.01$) with DPNN demonstrating a median MRR of 0.451 (median absolute deviation, MAD for short = 0.004) followed by the Vulcan method (median = 0.441, MAD = 0.005), KMNN (median = 0.444, MAD = 0.013), CFR (median = 0.444, MAD = 0.014), CBR (median = 0.442, MAD = 0.011), and LPAD (median = 0.44, MAD = 0.006). Recall that for MRR, the higher the better.

Similar results were encountered for the RPL metric. The DPNN model (median = 0.084, MAD = 0.006), was significantly superior to Vulcan (median = 0.097, MAD = 0.006), CFR (median = 0.091, MAD = 0.001), CBR (median = 0.09, MAD = 0.006), and LPAD (median = 0.091, MAD = 0.007), while no significant difference were found when comparing DPNN and KMNN (median = 0.084, MAD = 0.006), $p \leq 0.01$. Note that DPNN demonstrates better performance than KMNN, yet the difference is not statistically significant. Recall that for the RPL metric, lower numbers are preferable.

Table 3 summarizes the results and presents the performance of all methods also in specificity, sensitivity, positive predictive value (PPV) and negative predictive value (NPV) [53].

DPNN, KMNN and LPAD receive the number of clusters (or prototypes), $\ell$, as input. Since the data was originally generated from 5 prototypes, we chose to use the correct input for all three methods. A full sensitivity analysis by ranging $\ell$ from 2 to 9 shows that slight changes to

**Table 3. Experiment 1 (synthetic data): Median scores for each evaluated method (column) and evaluation metric (row).** For all metrics ecepts *RPL*, the higher the better. For *RPL*, the lower the better. Results in bold are significantly superior to non-bold results, $p \leq 0.01$.

| Method / Metric | DPNN | Vulcan | CFR | CBR | KMNN | LPAD |
|---|---|---|---|---|---|---|
| *MRR* | **0.451** | 0.441 | 0.444 | 0.442 | 0.444 | 0.44 |
| *RPL* | **0.084** | 0.097 | 0.091 | 0.09 | **0.084** | 0.091 |
| *AUC* | 0.751 | **0.777** | 0.754 | 0.755 | 0.739 | 0.732 |
| *Sensitivity* | 0.07 | 0.182 | **0.257** | 0.087 | 0.024 | 0.183 |
| *Specificity* | **0.986** | 0.964 | 0.914 | 0.808 | 0.98 | 0.824 |
| *PPV* | 0.502 | 0.488 | **0.622** | 0.366 | 0.471 | 0.375 |
| *NPV* | 0.333 | **0.344** | 0.319 | 0.314 | 0.327 | 0.329 |

For all metrics ecepts *RPL*, the higher the better. For *RPL*, the lower the better. Results in bold are significantly superior to non-bold results, $p \leq 0.01$.

this input result in little effective change for DPNN and that the correct number of clusters could be identified using the elbow method, see Appendix F for complete details.

## 5.4 Experiment 2: MDD clinical data

As discussed before, in real-world clinical data our evaluation possibilities are narrower since the data includes only the outcome of a single treatment with no practical way to obtain the counterfactuals. Therefore, the MRR and RPL metrics cannot be evaluated on real data and are replaced by the RR metric (see Section 5.2). Therefore in experiment 2 we report the results on the AUC and RR metrics alone.

**5.4.1 Data.**   The real world data consisted MDD patient level data from multiple clinical trials: CO-MED [54], STAR*D [55], REVAMP [56], EMBARC [57] and IRL-GREY [58]. Alltogether, the dataset described 4754 patients. In these studies, patients were and randomized to different anti-depressant treatments, and regularly reported their depressive symptoms every few days, with common questionnaires for evaluating depression, such as The 16-Item Quick Inventory of Depressive Symptomatology (QIDS) [59] and Hamilton Depression Rating Scale (HAM-D) [60]. Each patient was described by 26 features, including features describing the initial depressive disorders (e.g. suicidal ideation and fatigue) as reported in the questionnaires at the beginning of the study and social-demographic features (e.g. age and education). In addition, the data included for each patient a "treatment feature", describing the treatment he or she received. The "treatment feature" indicated one from 6 possible antidepressant treatment courses: escitalopram, citalopram, venlafaxine, sertraline, and the combinations of bupropion and escitalopram, and mirtazapine and venlafaxine. In addition, every patient had a binary outcome label: remission or no-remission. The full description of the data, including the 26 features and the pre-processing procedure are presented in Appendix B.

**5.4.2 Training.**   Unlike Experiment 1 where the true number of prototypes was known (chosen by us), in this experiment, we first evaluated the performance of each method by varying the value of $\ell$ from 2 to 9. The full sensitivity analysis results are presented in Appendix F. We found that in both DPNN and KMNN the model performed best with 6 prototypes ($\ell = 6$) and that the LPAD method performed best with three prototypes ($\ell = 3$). Therefore, in this experiment we set $\ell = 6$ for the DPNN and KMNN and $\ell = 3$ for the LPAD evaluation.

**5.4.3 Results.**   As was the case in Experiment 1, the results using both metrics (AUC and RR) were not normally distributed using the Shapiro-Wilk normality test [48]. As such, following the same analysis procedure of Experiment 1, we use the Friedman's test followed by posthoc Wilcoxon tests for pairwise comparisons with Bonferroni correction.

Using the Friedman's test, we found that the methods vary significantly in both RR and AUC scores ($p \leq 0.01$).

For the RR metric, pairwise comparisons reveal that the DPNN significantly outperforms all other benchmarks with a median score of 0.446 (MAD = 0.039) compared to Vulcan which demonstrated a median score of 0.413 (MAD = 0.035), CFR (median = 0.418, MAD = 0.045), CBR (median = 0.415, MAD = 0.025), KMNN (median = 0.412, MAD = 0.032) and LPAD (median = 0.411, MAD = 0.016), $p \leq 0.01$. Interestingly, the KMNN significantly outperformed all other benchmarks (other than DPNN), $p \leq 0.01$. It is important to note that the general remission rate, i.e. the portion of remission-labeled patients in the entire dataset, is only 0.355. Namely, all evaluated methods have demonstrated an expected added benefit compared to a random treatment selection procedure. Note that the RR metric is based on a varying number of samples (see Section 5.2). Nonetheless, the above results are found to be statistically significant. Table 4 summarizes the results. As mentioned above (Section 5.2), the RR metric is based only on a subset of patients from the test set (the patients who received the

**Table 4. Experiment 2 (clinical data): Median scores for each evaluated method (column) and evaluation metric (row).** For both metrics, the higher the better. Results in bold are significantly superior to non-bold results, $p \leq 0.01$.

| Metric | DPNN | Vulcan | CFR | CBR | KMNN | LPAD |
|---|---|---|---|---|---|---|
| RR | **0.446** | 0.413 | 0.418 | 0.415 | 0.412 | 0.411 |
| AUC | 0.64 | **0.65** | 0.626 | 0.58 | 0.595 | 0.608 |
| Sensitivity | 0.075 | 0.175 | **0.471** | 0.25 | 0.032 | 0.017 |
| Specificity | 0.965 | 0.926 | 0.817 | 0.811 | 0.987 | 0.985 |
| PPV | 0.522 | 0.509 | **0.611** | 0.385 | 0.492 | 0.343 |
| NPV | 0.36 | 0.37 | 0.306 | **0.365** | 0.357 | 0.355 |

For both metrics, the higher the better. Results in bold are significantly superior to non-bold results, $p \leq 0.01$.

optimal treatment within the test set). In all models, this group of users were in average between 30% and 40% of the test set (between 287 and 383 patients).

As was the case in Experiment 1, Vulcan was significantly superior to all other methods in terms of AUC achieving a median score of 0.65 (MAD = 0.001) compared to DPNN (median = 0.64, MAD = 0.011), CBR (median = 0.58, MAD = 0.013), KMNN (median = 0.595, MAD = 0.012), CFR (median = 0.626, MAD = 0.01), LPAD (median = 0.608, MAD = 0.011), $p \leq 0.01$. Nonetheless, DPNN significantly outperforms CFR, KMNN, CBR and LPAD methods, $p \leq 0.01$.

Table 4 summarizes the main results presented above. In addition, the table presents the performance of all methods in specificity, sensitivity, PPV and NPV measures.

## 6 Discussion

The experimental results presented in Section 5 demonstrate the advantages and limitations of our proposed DPNN approach. In both experimental setups, we see that the DPNN favorably compares with several state-of-the-art methods for treatment selection in terms of its low discordance with the unknown optimal treatment selection policy at a minimal expense in prediction accuracy. Specifically, for the MRR, RPL and RR metrics, we see that the DPNN significantly outperforms all benchmark methods. On the other hand, in order to achieve this advantage, as the DPNN is shown to have slight reduction in its performance in terms of AUC. From a practical perspective, it is claimed that "a prediction model is only as good as its resulting agent's performance" [61]. Adopting this viewpoint means that, in several decision-making environments, the DPNN's tradeoff of AUC performance for MRR, RPL and RR performance is worthwhile. This is because the purpose of the model is to increase the number of patients reaching remission; a significant increase in this metric has more clinical value than less than a percent difference in an accuracy metric. As such, we expect that the DPNN approach will prove more valuable as a clinical decision support technology compared to the evaluated state-of-the-art methods.

When presenting a new approach such as the DPNN, it is worthwhile discussing its limitations. First, as discussed above, DPNN has a small decrease in AUC performance to improve other, arguably more important, metrics. In environments where AUC is the prominent metric for evaluation, this may result in other methods (such as Vulcan) being preferred. We believe that in many medical decision-making environments, and specifically in MDD treatment selection, DPNN advantages supersede its limitations.

Second, DPNN requires the number of prototypes to be defined before execution. This is a standard limitation to most clustering-based algorithms and can be largely mitigated by adopting one of the many clustering analysis techniques commonly used for determining the

optimal number of clusters to use in a dataset [62]. In addition, as shown in Appendix F, DPNN is *not* very sensitive to the number of clusters in the data considered, which requires further investigation in order to determine the best way to interpret the clinical or biological meaning of the resulting clusters. Lastly, DPNN is built on several rather restrictive assumptions as outlined in Section 4. Specifically, it is unlikely that the assumption that samples (i.e., patients) can be divided into sub-groups which significantly differ in their reactions to different treatments will hold in all decision-making domains. While this theoretically limits the applicability of the DPNN approach, it appears that such settings are prominent, especially in the mental healthcare domain. Given the abundance of mental conditions, their prevalence in the general public and the great costs associated with them, DPNN can prove useful to a variety of high-impact treatment selection environments.

The architecture of our network, and specifically the prototypes learned by the network, can potentially increase the interpretability of the network's results, which is essential in an automated depression treatment selection system. Recent work demonstrates interpretability can drive physicians trust in an automated treatment selection system, which in turn can influence their use of the system's results [63]. This makes model interpretability a prime target for improving the physician-AI interaction. Therefore, in future work we plan to investigate how to explain the DPNN results to clinicians and how the network's results are perceived by both the clinicians and the patients.

## 7 Conclusions

In this article, we propose and evaluate a novel deep learning-based approach, DPNN, that simultaneously identifies sub-groups of patients as well as predicts personalized treatment outcomes. Our approach is shown to strike a delicate balance between the fully personalized paradigm (which ignores any possible clustering of patients) and the sub-grouping paradigm (which ignores individual differences within the groups) and favorably compares to existing benchmarks using synthetic and real-world clinical data.

Focusing on the important challenge of personalizing depression treatment, our approach demonstrates significant advantages over existing state-of-the-art methods. These advantages can potentially be translated into a significant reduction of the burden of depression in both the patient level and in the population-level and lead to a superior level of care.

As mentioned above, the actual remission rate in our dataset was 35.5%. Therefore, the DPNN method produced an 8% absolute and 23% relative improvement over random treatment allocation. In current practice, patients can be prescribed any of a number of treatments, with treatments considered equally effective at the population level, approximating a random assignment at scale. Therefore, these results are potentially clinically significant- given the large number of patients with MDD, a 23% improvement over current practice could mean potentially a very large number of patients reaching remission earlier and without needing to try multiple treatments.

We intend to extend this work in two directions: First we are currently preparing our system to be tested in a live clinical trail. This trial may shed new light on additional factors relating to the treatment selection process. For example, we expect that the interpretability of the model will be identified as an issue for further research (see [64] for a recent overview). We believe that the DPNN approach can enable physicians to gain insights through the learned prototypes and their "resemblance" to each individual patient. We plan to investigate this human-computer interaction issue in the future. Second, in this work, we primarily focused on the performance of our models in terms of *selected treatment efficacy*. As such, we intend to extend our analysis of the prototypes outputted by associating clinical meanings to each

**Table 5. Hyper-parameters of the DPNN model in both experiments.**

| hyper-parameter | meaning | value in experiment 1 | value in experiment 2 |
|---|---|---|---|
| $\ell$ | number of prototypes | 6 | 5 |
| hidden-layers in $e$ | number of hidden-layers and nodes in the auto-encoder | 12 nodes, single layer | 18 nodes, single layer |
| hidden-layers in $h$ | number of hidden-layers and nodes in the classification network | 12 nodes, single layer | 11 nodes, single layer |
| learning-rate | controls how quickly the model is adapted during training | 0.0001 | 0.0001 |
| epochs | number of epochs in training | 75 | 90 |
| batch size | number of training examples utilized in one iteration | 10 | 10 |
| $\lambda_1$ | auto-encoder loss ($L_2$) weight (Eq 9) | 0.01 | 0.01 |
| $\lambda_2$ | auto-encoder prototype variance loss ($L_3$) weight (Eq 9) | 0.05 | 0.06 |
| $\alpha$ | balance of the prototype variance loss (Eq 6) | 0.85 | 0.95 |

prototype. This additional analysis and interpretation is challenging yet it can prove very beneficial from a clinical perspective. Last, we plan to experiment with additional decision-making domains, both medical and non-medical, to better understand the advantages and limitations of our approach.

# 8 Appendices

## A Model training parameters

The DPNN method includes several hyper-parameters that can be configured and effect the model's performance. In this appendix, we present the values of the hyper-parameters used in both our experiments. The full list of hyperparameters are presented below in Table 5.

In both experiments, we set the hyper-parameters $\lambda_1$ to 0.01 $\lambda_2$ to 0.05, $\alpha$ to 0.85 and the learning rate to 0.0001. These parameters were found to be most useful using a simple grid-search.

Regarding the inner layers of the model, we found that the model performed best, in both experiments, with one inner layer in the auto-encoder component (one inner layer in the encoder and a symmetrical layer in the decoder). In the first experiment, the layer consisted 14 nodes and in the second experiment 18 nodes.In addition, in both experiments the classifier layers components included one hidden layer, In experiment 1, the layer consisted 16 nodes and in experiment 2 we used 11 nodes.

We obtained the hyper-parameters by an automated process that searched various combinations of parameters and found the combination that yielded the best results.

**Test size influence.** In our experiments we used a 5-fold validation, therefor the test size was 20% of the original dataset. In order to investigate the influence of the test size on the performance of the DPNN model with the MDD data, we tried splitting the data by 4 other possible train-test ratios: 5% (test ratio), 10%, 25%, 50%. For each of these ratios, we ran the model 10 times and measured the median RR measure and the AUC. In Table 6, we present the

**Table 6. Train-test ratio influence on the performance of the DPNN model with the MDD data, medians results.**

| test-train ratio | RR | AUC |
|---|---|---|
| 5: 95 | 0.375 | 0.645 |
| 10: 90 | 0.41 | 0.635 |
| 20: 80 | 0.446 | 0.64 |
| 25: 75 | 0.411 | 0.621 |
| 50: 50 | 0.433 | 0.631 |

median result. The results in the third row, for the 20:80 ratio, are the results in we obtained with the 5-fold cross validation, which we reported in our paper. This ratio gave the best result in *RR* measure, which is our prime objective.

## B The MDD data description

The real world data consists of MDD patient-level data from several major clinical trials: CO-MED [54], STAR*D [55], REVAMP [56], EMBARC [57] and IRL-GREY [58]. Combining studies was necessary in order to include enough different treatments to produce a model of potential clinical value (as a model which simply assists in selecting between one treatment or another is not likely to be of clinical use when over a dozen treatments and treatment combinations can be selected from). CO-MED enrolled outpatients with MDD who were randomized to three anti-depressant treatment arms: escitalopram and placebo, bupropion and escitalopram, or mirtazapine and venlafaxine. The purpose of the trial was to assess whether combination treatment was superior to monotherapy, but similar remission rates were observed in each arm. STAR*D is the largest pragmatic trial of depression treatment ever conducted to date and followed patients through multiple courses of treatment; in our dataset we look at the first stage of treatment of the four levels of the study (which is the level analyzed in this study), all patients received citalopram, and the remission rate was 33%. we used the first stage because we were interested in predicting response to an initial monotherapy trial, and also because sample size decreased significantly in later stages of the study. As it was a pragmatic study, aside from requiring that patients have at least moderate severity major depression, there were few strict exclusion criteria (other than medical instability or substance abuse disorders that requiring detoxification, eating disorders, or obsessive compulsive disorders) making the study fairly representative of a real clinical sample (n = 2876). REVAMP was a study comparing a medication with two different psychotherapies added to the medication; we analyzed the medication only group, which included patients on escitalopram, bupropion, venlafaxine or mirtazapine (however due to small sample size we were unable to include the patients on mirtazapine for this analysis). Patients needed to have chronic depression, and were not allowed to have psychotic disorders, bipolar disorder, post traumatic stress disorder, obsessive compulsive disorder, eating disorders, substance abuse or personality disorders. EMBARC was a study focused on finding biomarkers of depression treatment response; we use the sertraline and bupropion arms of the study as these treatments as monotherapy were not available in other data sets we had access to. Finally, IRL-GREY was a study of older adults with depression; we use the data from the first stage of the study, where all patients recieved venlafaxine monotherapy.

All in all, we had seven different treatments available for the model to learn to predict remission rates for: bupropion, escitalopram, citalopram, venlafaxine, sertraline, and the combinations of bupropion and escitalopram, and mirtazapine and venlafaxine. However, in this study we excluded the group of patients that received bupropion only since it was very small in comparison to the other groups (65 patients). These are all first-line or combinations of first-line treatments and are commonly used in clinical practice [65]; while they are essentially equally effective at the population level when looking at the data from each of these studies, clinically they are used differently and in usual practice are thought to help different kinds of patients in a differential manner [66]. We did not use data from placebo arms from any of the studies; given these are all known to be effective treatments, comparison to placebo to prove efficacy was not necessary and would not have provided useful information in predicting differential treatment response, though in future work we plan to assess prediction of differential response to placebo, as in [67]. When determining patient remission status, we took an "intent

to treat" approach, using all patients (even those who dropped out before study end as long as they had been in the study for a minimum of two weeks, as prior to two weeks there is not likely to be any effect of treatment) and ascertaining remission at the last possible measurement in order to capture the state of the patient as they were leaving the study (as a parallel to the last point at which they would be leaving treatment in a real clinical setting). While this results in a remission rate that would be lower than that observed if one were to look only at remission rates at the end of the study and therefore to greater lack of balance between the remission and non-remission categories- it also provides a dataset closer to true clinical reality and which matches the remission rates reported in the included studies. The sample was 37.6% female. The sample overall included patients with similar remission rates, a high rate of chronic or recurrent depression, and generally with patients with psychiatric comorbidities permitted to enroll in most studies. All the datasets together included 4754 patients, with an overall 35.5 percent remission rate. These studies were ideal for our analysis because they included similar eligibility criteria and outcome measures and generally allowed clinicians to tailor patient dose to patient need and tolerance, as in real practice. They were all carried out as investigator-initiated studies of well-known treatments as well, and were more focused on comparative efficacy than on getting a new treatment approved.

The data initially included 213 features, and the model performed poorly on the the raw data. Therefore, we first employed the following methods for feature reduction in order to improve the various models' results:

1. Feature importance thresholding via randomized Lasso: This procedure randomly shuffled the samples and selects the set of features most closely linked to the label we were trying to predict (i.e. remission).

2. Recursive feature elimination with cross validation (RFECV): This method trains a model with subsets of the original feature list and detects the features with the most value for the performance of the model.

We were able to identify a list of 26 optimal parameters (not including the target value and treatment feature) for our feature selection by iteratively running the features with our models and assessing the prediction metrics. These methods were implemented from the Python package Sci-kit Learn.

In addition, at a later step during model development we removed one of the seven original treatment courses, bupropion, since it only included 63 instances in the data. We are currently working to secure more data on patients using this treatment, so future iterations of the model can include this and potentially other treatments as well.

**Features list.** Here we list the final features used in our model. all features are categorical and were reported by various questionnaires.

1. From HAM-D questionnaire: Poor appetite or overeating.

2. From HAM-D questionnaire: Impact of your family and friends.

3. From HAM-D questionnaire: Early morning insomnia.

4. From HAM-D questionnaire: Energy/ fatigability.

5. From HAM-D questionnaire: Pleasure/enjoyment.

6. From HAM-D questionnaire: Reactivity of mood.

7. From HAM-D questionnaire: Suicidal ideation.

8. From HAM-D questionnaire: Sympathetic arousal.

9. From HAM-D questionnaire: Outlook towards future.

10. From HAM-D questionnaire: Fatigue.

11. From HAM-D questionnaire: Has private insurance.

12. QIDS total score (the sum of all symptom values in the QIDS questionnaire).

13. Number of relatives living with patient.

14. From QIDS questionnaire: Appetite (increased).

15. From QIDS questionnaire: Concentration/decision making.

16. From QIDS questionnaire: Energy/fatigability.

17. From QIDS questionnaire: Involvement.

18. From QIDS questionnaire: Mood (sadness).

19. From QIDS questionnaire: Mid-nocturnal insomnia.

20. From QIDS questionnaire: Suicidal ideation.

21. Climbing several flights of stairs.

22. Currently employed partial or full time.

23. 16 or more years of education.

24. Number of friends living with patient.

25. One to four number of relatives living with patient.

26. One to five number of persons in household.

## C Synthetic data generation

In this appendix we describe in detail the process of data generation for the synthetic data we used in Experiment 1 (Section 5.3). First, we randomly created a set of 5 prototypes ($\ell = 5$), where each prototype is represented as a 10 feature vector ($q = 10$), each of which is sampled from a normal distribution with a mean of 0 and standard deviation of 10. For each prototype, we generated a set of 2,000 fictitious patients *in the latent-space* according to the same normal distribution around the prototypes. Each patient was randomly assigned to one of four treatments ($k = 4$).

Each of the four treatments was associated with a randomly chosen non-linear *remission function* that maps patients' latent features to a probability of remission. The remission function determines the remission probability using two linear matrix multiplications with a RELU activation function in between. All the values of the matrices were randomly generated from a standard normal distribution. The first matrix's size was $10 \times 5$ and the second matrix $5 \times 2$. As before, slight variations to these parameters and other choices of non-trivial remission functions have demonstrated similar results. The outcome for each patient-treatment pair, namely remission or non-remission, was chosen using a softmax operator over the results of the above calculation.

To complete the process, each generated patient was "decoded" into 16 observable (non-latent) features using a decoder function. The decoder function uses a matrix multiplication

where the matrix's size was $10 \times 16$. All the values of the matrix were randomly generated from a standard normal distribution. In addition, in order to avoid over-simplicity, the decoder function added 4 irrelevant features to the observed space. These features that were randomly generated from the standard normal distribution. altogether, each patient was now represented by 20 features ($d = 20$).

Overall, the above process produced a synthetic dataset of 10,000 patients represented in the non-latent space, each associated with one random treatment and an outcome. The script we used for this process was implemented in python and is publicly available at: https://github.com/Aifred-Health/DPNN_Experiment/tree/master/Experiment/Synthetic_Data_Experiment.

## D LPAD

In our experiments we compared our model to LPAD: a sub-grouping method for treatment of depression, based on Latent Profile Analysis. We mentioned above (Section 5) that this method we first divided our dataset into train and test sets (80%-20% split), and executed the original LPAD procedure on the training set. Then, using the test set, we first classified each patient to the nearest cluster. We then use the data from the patients allocated to each cluster in order to estimate the remission likelihood for that cluster through a standard maximum likelihood estimation. This way, we were able to obtain the optimal treatment for each patient, based on his associated cluster. We used this procedure in order to calculate all the metrics described in Section 5.2.

## E CFRNet adjustment

CFRnet was originally designed for data including two groups of treatments, however the authors mention that the method can be trivially extended to multiple treatments, by estimating the outcome for each pair of treatments and aggregating the results [31]. In our evaluation we followed their suggestion in both experiments as follows: For each sample, we predicted the outcomes of each possible per of interventions (treatments), and than we averaged the results for each intervention. Consequently, we obtained for each sample a predicted outcome for each possible intervention and specifically for the intervention he or she actually received, and we used these results for calculating the various metrics.

## F Sensitivity analysis

In our experiments we evaluated three methods that involve prototyping or clustering: DPNN, KMNN and LPAD. Before comparing the models' performance, we first analyzed the effect of

**Table 7. Sensitivity analysis- synthetic data.** Performance by Number of Clusters.

| | DPNN | | | kmean | | | LPAD | | |
|---|---|---|---|---|---|---|---|---|---|
| | AUC | MRR | RPL | AUC | MRR | RPL | AUC | MRR | RPL |
| 2 | 0.73 | 0.42 | 0.1 | 0.71 | 0.44 | 0.09 | 0.74 | 0.44 | 0.08 |
| 3 | 0.74 | 0.44 | 0.1 | 0.71 | 0.44 | 0.11 | 0.74 | 0.44 | 0.08 |
| 4 | 0.74 | 0.44 | 0.1 | 0.76 | 0.44 | 0.08 | 0.74 | 0.44 | 0.09 |
| 5 | 0.75 | 0.45 | 0.09 | 0.74 | 0.44 | 0.09 | 0.73 | 0.44 | 0.09 |
| 6 | 0.75 | 0.45 | 0.09 | 0.76 | 0.42 | 0.08 | 0.74 | 0.44 | 0.09 |
| 7 | 0.73 | 0.44 | 0.1 | 0.77 | 0.44 | 0.09 | 0.74 | 0.44 | 0.09 |
| 8 | 0.77 | 0.43 | 0.08 | 0.77 | 0.41 | 0.08 | 0.74 | 0.44 | 0.1 |
| 9 | 0.75 | 0.44 | 0.09 | 0.76 | 0.46 | 0.08 | 0.73 | 0.43 | 0.1 |

**Table 8. Sensitivity analysis- MDD data.** Performance by Number of Clusters.

| | DPNN | | Kmean | | LPAD | |
|---|---|---|---|---|---|---|
| | **AUC** | **RR** | **AUC** | **RR** | **AUC** | **RR** |
| 2 | 0.61 | 0.42 | 0.59 | 0.41 | 0.6 | 0.42 |
| 3 | 0.6 | 0.42 | 0.6 | 0.38 | 0.61 | 0.42 |
| 4 | 0.61 | 0.43 | 0.59 | 0.42 | 0.61 | 0.39 |
| 5 | 0.64 | 0.43 | 0.59 | 0.42 | 0.61 | 0.4 |
| 6 | 0.64 | 0.43 | 0.59 | 0.42 | 0.62 | 0.39 |
| 7 | 0.62 | 0.42 | 0.63 | 0.38 | 0.62 | 0.39 |
| 8 | 0.63 | 0.43 | 0.63 | 0.36 | 0.61 | 0.4 |
| 9 | 0.64 | 0.42 | 0.63 | 0.39 | 0.62 | 0.4 |

the number of prototypes (or clusters) on the performance of the models. All three methods were executed with the number of prototypes ranging from 2 to 9. For each number of prototypes, we ran the models 5 times and obtained the average value of all metrics. We found that, in Experiment 2, for the RR metric in the MDD data, the model performed best with 6 prototypes, for both the *DPNN* and *KMNN* models. The results for both experiments, are presented in Tables 7 and 8.

## Acknowledgments

Aifred Health-affiliated co-authors collaborated with other co-authors in the conduct of this work. DB, CA, RF, JM are employees and officers of Aifred Health. AK, AR, CA, JM, RF, and DB are co-inventors on a patent pending relating to this work.

## Author Contributions

**Conceptualization:** Akiva Kleinerman, Ariel Rosenfeld, Amit Yaniv-Rosenfeld.

**Data curation:** Akiva Kleinerman, Joseph Mehltretter, Jordan Karp, Charles F. Reynolds.

**Formal analysis:** Akiva Kleinerman.

**Investigation:** Akiva Kleinerman, David Benrimoh, Robert Fratila, Caitrin Armstrong, Joseph Mehltretter, Eliyahu Shneider, Gustavo Turecki, Adam Kapelner.

**Methodology:** Akiva Kleinerman, David Benrimoh, Robert Fratila, Caitrin Armstrong, Amit Yaniv-Rosenfeld.

**Project administration:** Ariel Rosenfeld.

**Resources:** Akiva Kleinerman, Ariel Rosenfeld.

**Validation:** Akiva Kleinerman, David Benrimoh, Eliyahu Shneider.

**Visualization:** Akiva Kleinerman.

**Writing – original draft:** Ariel Rosenfeld, David Benrimoh.

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
