## [Decision Letter · Decision Letter 0]

21 Jul 2021

PONE-D-21-18970

Treatment selection using prototyping in latent-space with application to depression treatment

PLOS ONE

Dear Dr. Kleinerman,

Thank you for submitting your manuscript to PLOS ONE. After careful consideration, we feel that it has merit but does not fully meet PLOS ONE’s publication criteria as it currently stands. Therefore, we invite you to submit a revised version of the manuscript that addresses the points raised during the review process.

In particular:

Address the issue of non-compliance with PLOS data availability policy (in particular, the authors must provide compelling reasons if the data is not shared publicly; the point of contact for requesting the data is not stated and should not be the authors, see FAQs at the address above).Revise the language to improve the flow and readability of the manuscript.Clarify whether the github repository with the code is meant to be released upon acceptance or kept private.Provide sufficient details to reproduce the results (ideally also releasing the source code publicly; in any case the paper should contain enough information to understand the methods even without looking at the code).Improve the statistical analysis of the results, as suggested by the reviewers (3 and 4 in particular).

We look forward to receiving your revised manuscript.

Kind regards,

Luca Citi, PhD

Academic Editor

PLOS ONE

2. Thank you for stating the following in the Acknowledgments Section of your manuscript: [This study was supported by the Chief Scientist Office, Israeli Ministry of Health 697 (CSO-MOH, IL) as part of grant #3-000015730 within Era-PerMed. DB, CA, RF and 698 JM are shareholders or employees of Aifred Health. This work was supported by an 699Era-PerMed 2020 Grant.]

 [AK and AR  were supported by the Chief Scientist Office, Israeli Ministry of Health (CSO-MOH, IL  url: https://www.health.gov.il/English/Pages/HomePage.aspx) as part of grant #3-000015730 within Era-PerMed. The funders had no role in study design, data collection and analysis, decision to publish, or preparation of the manuscript.

AK and AR have received honoraria from Aifred Health (https://www.aifredhealth.com/).]

Additionally, because some of your funding information pertains to commercial funding, we ask you to provide an updated Competing Interests statement, declaring all sources of commercial funding. 

In your Competing Interests statement, please confirm that your commercial funding does not alter your adherence to PLOS ONE Editorial policies and criteria by including the following statement: "This does not alter our adherence to PLOS ONE policies on sharing data and materials.” as detailed online in our guide for authors http://journals.plos.org/plosone/s/competing-interests. If this statement is not true and your adherence to PLOS policies on sharing data and materials is altered, please explain how. Please include the updated Competing Interests Statement and Funding Statement in your cover letter. We will change the online submission form on your behalf.

3. Thank you for providing the following Funding Statement:  

[AK and AR  were supported by the Chief Scientist Office, Israeli Ministry of Health (CSO-MOH, IL  url: https://www.health.gov.il/English/Pages/HomePage.aspx) as part of grant #3-000015730 within Era-PerMed.The funders had no role in study design, data collection and analysis, decision to publish, or preparation of the manuscript.

AK and AR have received honoraria from Aifred Health (https://www.aifredhealth.com/). ]. 

We note that one or more of the authors is affiliated with the funding organization, indicating the funder may have had some role in the design, data collection, analysis or preparation of your manuscript for publication; in other words, the funder played an indirect role through the participation of the co-authors. 

If the funding organization did not play a role in the study design, data collection and analysis, decision to publish, or preparation of the manuscript and only provided financial support in the form of authors' salaries and/or research materials, please review your statements relating to the author contributions, and ensure you have specifically and accurately indicated the roles that these authors had in your study in the Author Contributions section of the online submission form. Please make any necessary amendments directly within this section of the online submission form.  Please also update your Funding Statement to include the following statement: “The funder provided support in the form of salaries for authors [insert relevant initials], but did not have any additional role in the study design, data collection and analysis, decision to publish, or preparation of the manuscript. The specific roles of these authors are articulated in the ‘author contributions’ section.

If the funding organization did have an additional role, please state and explain that role within your Funding Statement. 

Please also provide an updated Competing Interests Statement declaring this commercial affiliation along with any other relevant declarations relating to employment, consultancy, patents, products in development, or marketed products, etc.  

Reviewers' comments:

Reviewer's Responses to Questions

**Comments to the Author**

1. Is the manuscript technically sound, and do the data support the conclusions?

Reviewer #1: Yes

Reviewer #2: Yes

Reviewer #3: Partly

Reviewer #4: Yes

2. Has the statistical analysis been performed appropriately and rigorously? 

Reviewer #1: Yes

Reviewer #2: Yes

Reviewer #3: No

Reviewer #4: Yes

3. Have the authors made all data underlying the findings in their manuscript fully available?

Reviewer #1: Yes

Reviewer #2: No

Reviewer #3: No

Reviewer #4: Yes

4. Is the manuscript presented in an intelligible fashion and written in standard English?

Reviewer #1: Yes

Reviewer #2: No

Reviewer #3: Yes

Reviewer #4: No

5. Review Comments to the Author

Reviewer #1: The full meaning of KMNN is missing on page 6.

section 4

line 4 "to" was repeated

the equations from page 7 should be label for easy understanding

How do you intend to obtain an optimal value of k for the K-Means clustering and the KNN?

page 11 under section 5.1 line 14 "mesure" wrong spelling

List the hyperparameters used in this study with reasons

Explain the neural network architecture used in this study

Reviewer #2: The concept of providing clinicians with tools to help them plan, identify and assign the right treatment that optimizes gains and fits the patient's needs is of great importance. In precision medicine, one of these tools is deep learning systems which can integrate and model heterogeneous data from an individual patient, allowing better predictions and recommending treatment options tailored to each patient's individual characteristics and needs. In this manuscript authors propose a deep learning-based approach for automated treatment selection for patients with Major Depressive Disorder. According to their description, their method is based on a neural network that (a) identifies sub-groups of patients (prototypes) in the latent space that differ in their characteristics and their expected responses to the available courses of treatment and (b) approximates outcome prediction in a personalized manner by predicting the remission probability for each patient-treatment pair based on their resemblances to identified sub-groups. The model outputs are both the group membership for a given patient and the personalized probabilities of treatment success for each possible treatment.

The work presenting in this paper is interesting and authors adequately describe how their study differs from previous work. They further appropriately discuss their findings and clearly state the limitations of their approach alongside with the future work. Their method is well-explained and the authors have provided within the document information which seems enough to allow others to validate their study. My main questions have been already answered by the authors in the discussion section. However, the data are not available for review since the authors do not provide access to their data and the link into the github, where their implementation and the benchmarks are, does not work. In addition, while the study appears to be sound, the language in some parts is unclear and making it difficult to follow. For example, the third paragraph in Introduction section, there are two consecutive sentences that begin with “however” (which is also used 3 times in the same paragraph). Similarly, in Section 4, the repetition of “however, recall” and in general, asking readers to recall 8 times within the document is becoming very tiring. I would suggest, the authors to revise the language to improve the flow and readability of the text (especially in the middle sections). The citations are also in the wrong order and the equations are not all numbered which would be helpful for other authors who would like to refer and use them. Overall, at its present form, with the minor improvements suggested, the manuscript could make an acceptable case for publication.

Reviewer #3: While the topic is of interest, the manuscript is not well written. The description of the methodology and results is quite confusing and hard to follow. Moreover, the study is not reproducible because of lack of details - it is quite surprising to read a whole manuscript and have no idea which features have been used to classify MDD patients!. Authors should make quite a significant effort to improve on the manuscript readability and provide more information about the synthetic and real data that was used. Other detailed comments follow below.

authors should carefully check the manuscript for typos (e.g., "performed poorly on the the")

data availability statement does not meet the Plos criteria (reason why data is not open is not mentioned; the ethics committee eventually limiting that is not mentioned either. Which contact email has to be used to request the data? )

abstract is quite generic and does not provide quantitative results. Also, the data that is taken as input should be mentioned as well.

The Introduction does not mention either the data that the proposed network takes as input.

In the incidence of MDD in Sec 2.2, the year to which the cases refers to should be mentioned

"It is important to note that MDD is a brain illness" this statement should be avoided as it has been demonstrated that MDD involved much more that brain areas, including peripheral nervous branches.

In introduction the clinical state of the art for MDD treatment, DSM-V criteria should be mentioned

I don't get what's in lines 279-280 if Y is binary

Section 5.3.1 is quite unclear and does not allow to reproduce the results. Much more info should be provided in order to make the study reproducible. Once again, which kind of data has been used and taken as input?????

a total of 10 cross-fold validation steps seem quite low and should be increased significantly.

When using non-parametric statistical tests, median and median absolute deviations should be used for consistency in place of mean and std. Incidentally, results in 5.3.3 are quite difficult to read, may be box-plots help.

Authors should show standard performance metrics as specificity and sensitivity, as well as positive/negative predictive values, considering the classes remission/not remission

Reviewer #4: In this paper, the authors propose a model to learn subgroups of patients for depression treatment recommendations. The proposed model is constructed based on neural network architecture. It aims to learn subgroups that have heterogeneous treatment effects based on patients' features extracted in the latent space. Experiments on simulated and real data were conducted.

Detailed comments:

The Introduction Section is a bit repetitive with the Section of Related Work. The authors can make the contents more concise.

In the simulation studies, Section 5.3.2 line 533, “We repeated this process 10 times”, “All together, we obtained 50 samples of each metric for each method.” More replications are needed. Also, for each replication of cross-validation, one value should be reported over all validation sets instead of reporting 5 values.

For the simulations, it would be good to conduct the same experiment under different training sample sizes to see whether the algorithm converges to the truth with the increase of the sample size. And it would be good to give the readers an idea what sample size is needed for a decent performance.

I wonder in the simulation studies, given the correct number of subgroups, can the true subgroup compositions be recovered by the algorithm?

Can the authors provide more computational details, such as learning rate, number of epochs, computational time, etc.?

In the real data application study, to inform the readers, can the authors list the final variables that were selected for running the algorithm?

In line 320 and 323, the notation for the number of prototypes is not consistent, $\\mathcal l$ and L in $p_L$, $d_L$.

In appendix C, line 1042, should the decoder matrix be of dimension 10 * 20?

6. PLOS authors have the option to publish the peer review history of their article (what does this mean?). If published, this will include your full peer review and any attached files.

Reviewer #1: **Yes: **Stephen Gbenga Fashoto

Reviewer #2: No

Reviewer #3: No

Reviewer #4: No

---

## [Author Response · Author response to Decision Letter 0]

31 Aug 2021

We are submitting a revision of our earlier submission.

We wish to thank the reviewers for their insightful comments.

The text below explains in detail our responses to the comments of the reviewers in the decision letter for our earlier version. We marked the changes in the text in blue.

Please note that in order to best respond to reviewer comments regarding our description of the psychiatric data, we opted to consult with two collaborators who provided us data from their study for model training. Their assistance was key in the text revisions and in helping us improve the abstract and presentation of the clinical results; as such, we have added them to the paper as co-authors in recognition of this assistance. 

Best regards,

Akiva Kleinerman on behalf of all co-authors.

 

PLOS ONE Editor’s Comments

● Address the issue of non-compliance with PLOS data availability policy (in particular, the authors must provide compelling reasons if the data is not shared publicly; the point of contact for requesting the data is not stated and should not be the authors, see FAQs at the address above).

Response: We apologize for this oversight when submitting originally; the correct data availability statement and the sources of data, and how to request them from their owners, is now updated. 

● Revise the language to improve the flow and readability of the manuscript.

Response: This has been done. We revised the language carefully and made changes in order to improve the readability of the manuscript. 

● Clarify whether the GitHub repository with the code is meant to be released upon acceptance or kept private.

Response: The GitHub repository was meant to be public; it was an error that it was not. It is now public, our apologies. The link for the public repository is: https://github.com/Aifred-Health/DPNN_Experiment

● Provide sufficient details to reproduce the results (ideally also releasing the source code publicly; in any case the paper should contain enough information to understand the methods even without looking at the code).

Response: We have provided this information as part of responses to reviewer comments. 

● Improve the statistical analysis of the results, as suggested by the reviewers (3 and 4 in particular).

Response: Thank you. This has been thoroughly done as per our response to reviewers. 

 

The Reviewers’ Comments 

Reviewer #1:

Comment: The full meaning of KMNN is missing on page 6.

Response: We thank the reviewer for this comment. We agree that the definition of KMNN at page 6 should be better explained, we added more details in the text. 

Comment: section 4 line 4 "to" was repeated

Response: Fixed, thanks.

Comment: the equations from page 7 should be label for easy understanding

Response: added, thanks.

Comment: How do you intend to obtain an optimal value of k for the K-Means clustering and the KNN?

Response: We ran the model with all possible k values between 2 and 9 and found that KMNN performed best with six clusters, we presented the results in Appendix F. 

Comment: page 11 under section 5.1 line 14 "mesure" wrong spelling

Response: Fixed, thanks.

Comment: List the hyperparameters used in this study with reasons

Response: We obtained the hyper-parameters by an automated process that searched various combinations of parameters and found the combination that yielded the best results. This is now better clarified in the revised text.

Comment: Explain the neural network architecture used in this study

Response : We have now amended and expanded the explanation of the neural net, in Section 4, so that it will be more clear. 

Reviewer #2: 

Comment: The concept of providing clinicians with tools to help them plan, identify and assign the right treatment that optimizes gains and fits the patient's needs is of great importance. In precision medicine, one of these tools is deep learning systems which can integrate and model heterogeneous data from an individual patient, allowing better predictions and recommending treatment options tailored to each patient's individual characteristics and needs. In this manuscript authors propose a deep learning-based approach for automated treatment selection for patients with Major Depressive Disorder. According to their description, their method is based on a neural network that (a) identifies sub-groups of patients (prototypes) in the latent space that differ in their characteristics and their expected responses to the available courses of treatment and (b) approximates outcome prediction in a personalized manner by predicting the remission probability for each patient-treatment pair based on their resemblances to identified sub-groups. The model outputs are both the group membership for a given patient and the personalized probabilities of treatment success for each possible treatment.

The work presenting in this paper is interesting and authors adequately describe how their study differs from previous work. They further appropriately discuss their findings and clearly state the limitations of their approach alongside with the future work. Their method is well-explained and the authors have provided within the document information which seems enough to allow others to validate their study. My main questions have been already answered by the authors in the discussion section. However, the data are not available for review since the authors do not provide access to their data and the link into the github, where their implementation and the benchmarks are, does not work. In addition, while the study appears to be sound, the language in some parts is unclear and making it difficult to follow. For example, the third paragraph in Introduction section, there are two consecutive sentences that begin with “however” (which is also used 3 times in the same paragraph). Similarly, in Section 4, the repetition of “however, recall” and in general, asking readers to recall 8 times within the document is becoming very tiring. I would suggest, the authors to revise the language to improve the flow and readability of the text (especially in the middle sections). The citations are also in the wrong order and the equations are not all numbered which would be helpful for other authors who would like to refer and use them. Overall, at its present form, with the minor improvements suggested, the manuscript could make an acceptable case for publication.

Response: We thank the reviewer for these important comments and feedback. We revised the paper carefully to improve the language and the flow. We also reorganized the citations so that they will be sorted according to their order of appearance. In addition, we clarified how the data can be accessed. We do not own this data, but all of it can be accessed upon request from its sources (the NIMH and the University of Pittsburgh). 

Reviewer #3:

Comment: While the topic is of interest, the manuscript is not well written. The description of the methodology and results is quite confusing and hard to follow. Moreover, the study is not reproducible because of lack of details - it is quite surprising to read a whole manuscript and have no idea which features have been used to classify MDD patients!. Authors should make quite a significant effort to improve on the manuscript readability and provide more information about the synthetic and real data that was used. Other detailed comments follow below.

Response: Thank you for this comment. We revised the paper and rewrote many paragraphs. Specifically, we added more information about the dataset, including examples of features (see Section 5.4.1), and more details about the neural networks In addition, we added the full list of features in Appendix B and referenced this addition in the manuscript.

Comment: authors should carefully check the manuscript for typos (e.g., "performed poorly on the the")

Response: We made major revisions in order to improve the language and the flow. Specifically, we rewrote and reorganized the introduction and the related work. 

Comment: data availability statement does not meet the Plos criteria (reason why data is not open is not mentioned; the ethics committee eventually limiting that is not mentioned either. Which contact email has to be used to request the data? )

Response: Data used for this study does not belong to the Authors and they are not legally permitted to share it. As such we have updated the data availability statement as follows: “The Data was provided from two sources. The first is the dataset of the IRL-GREY study and those interested in using this data should contact the Institutional Review Board of the University of Pittsburgh at askirb@pitt.edu. The other data source was the NIMH: Data and/or research tools used in the preparation of this manuscript were obtained from the National Institute of Mental Health (NIMH) Data Archive (NDA). NDA is a collaborative informatics system created by the National Institutes of Health to provide a national resource to support and accelerate research in mental health. Dataset identifier:10.15154/1523049. This manuscript reflects the views of the authors and may not reflect the opinions or views of the NIH or of the Submitters submitting original data to NDA. Those wishing to use this data can make a request to the NIMH (visit https://nda.nih.gov/).” 

Comment: abstract is quite generic and does not provide quantitative results. Also, the data that is taken as input should be mentioned as well.

Response: thanks for this important comment, we added a sentence in the abstract that provides quantitative results and added more details about the data.

Comment: The Introduction does not mention either the data that the proposed network takes as input.

Response: We added to the introduction more details regarding the MDD dataset and referenced both the paragraph in the evaluation Section (5.4.1) and the Appendix (B)

Comment: In the incidence of MDD in Sec 2.2, the year to which the cases refers to should be mentioned

Response: We thank the reviewer for this comment. We added the relevant years in the paragraph

Comment: "It is important to note that MDD is a brain illness" this statement should be avoided as it has been demonstrated that MDD involved much more that brain areas, including peripheral nervous branches.

Response: We thank the reviewer for this pointing our the problem in this sentence, we rephrased the sentence so that it will not imply that MDD only involves the brain. 

Comment: In introduction the clinical state of the art for MDD treatment, DSM-V criteria should be mentioned

Response: We added a paragraph in Section 2.2 that mentions the DSM-V criteria. We wrote: “MDD is diagnosed based on a heterogenous group of symptoms, and two patients with depression can have very different clinical phenomenology. The DSM-5 (APA, 2013) criteria for depression include loss of interest or pleasure in usual activities, depressed mood, increased or decreased appetite or weight, increased or decreased psychomotor activity, increased or decreased sleep, fatigue, poor concentration, feelings of guilt or worthlessness, and suicidality; to be diagnosed with major depression a patient must have at least five of these symptoms and at least one of these must be depressed mood or reduced interest or pleasure.” 

Comment: I don't get what's in lines 279-280 if Y is binary 

Response: We thank the reviewer for this comment. We meant that there are k binary outcomes, one for each possible treatment. Indeed, this was not clear enough in the text and therefore we now clarified this in the revised text.

Comment: Section 5.3.1 is quite unclear and does not allow to reproduce the results. Much more info should be provided in order to make the study reproducible. Once again, which kind of data has been used and taken as input?????

Response: We uploaded the script we used in order to generate the synthetic data and added more detailed information regarding the process in Appendix C. I addition we added more information regarding the Depression Dataset in Section 5.4.1, the introduction and in Appendix 2.

Comment: a total of 10 cross-fold validation steps seem quite low and should be increased significantly.

Response: We thank the reviewer for this comment. We added 90 cross-fold validation (100 in total) and averaged all the results, we updated the results accordingly. 

Comment: When using non-parametric statistical tests, median and median absolute deviations should be used for consistency in place of mean and std. Incidentally, results in 5.3.3 are quite difficult to read, may be box-plots help.

Response: We added the results of the medians and the median absolute deviations in an Appendix. The values of the medians are very close to the means, in all evaluated methods, and the means are normally distributed. 

Comment: Authors should show standard performance metrics as specificity and sensitivity, as well as positive/negative predictive values, considering the classes remission/not remission. 

Response: We added the performance metrics of specificity and sensitivity of all methods in both Experiments. We added the results to the tables in subsections 5.3.3 and 5.4.3 

Reviewer #4: 

In this paper, the authors propose a model to learn subgroups of patients for depression treatment recommendations. The proposed model is constructed based on neural network architecture. It aims to learn subgroups that have heterogeneous treatment effects based on patients' features extracted in the latent space. Experiments on simulated and real data were conducted.

Detailed comments:

Comment: The Introduction Section is a bit repetitive with the Section of Related Work. The authors can make the contents more concise.

Response: We rewrote and reorganized the introduction and the related work. Specifically, we tried to get rid of the repetitive parts and to make the Section more concise. 

Comment: In the simulation studies, Section 5.3.2 line 533, “We repeated this process 10 times”, “All together, we obtained 50 samples of each metric for each method.” More replications are needed. Also, for each replication of cross-validation, one value should be reported over all validation sets instead of reporting 5 values.

Response: We thank the reviewer for this important comment. We added 90 cross-fold validation (100 in total) and reported a single value for each cross validation. We averaged the results and updated the results accordingly. 

Comment: For the simulations, it would be good to conduct the same experiment under different training sample sizes to see whether the algorithm converges to the truth with the increase of the sample size. And it would be good to give the readers an idea what sample size is needed for a decent performance.

Response: Thank you for this comment. We now performed more experiments to investigate the influence of the test size on the performance of the DPNN model: we tried splitting the data various ratios of train and test sets, we added the results in Appendix A. We found that the model performed best with a 20 percent test size. 

Comment: I wonder in the simulation studies, given the correct number of subgroups, can the true subgroup compositions be recovered by the algorithm?

Response: Yes, we found that our algorithm (DPNN) performs best when it is aimed to find the exact number of prototypes that were used to generate the data. In Appendix F (Sensitivity Analysis) we provide the results of our network for various values of the number of prototypes. We can observe that in the synthetic data the network performs best when aimed to find 5 prototypes, which is the number of prototypes used for generating the data. 

Comment: Can the authors provide more computational details, such as learning rate, number of epochs, computational time, etc.?

Response: Yes, we added more computational details in Subsection 5.3.2 and added a full list of hyper parameters including learning rate, batch size and epochs. 

Comment: In the real data application study, to inform the readers, can the authors list the final variables that were selected for running the algorithm?

Response: Yes, we now added full list of variables to Appendix B and referenced this addition in Section 5.3.1 

Comment: In line 320 and 323, the notation for the number of prototypes is not consistent, $\\mathcal l$ and L in $p_L$, $d_L$.

Response: Fixed, thanks. 

Comment: In appendix C, line 1042, should the decoder matrix be of dimension 10 * 20? 

Response: Thanks for this comment. The decoded samples included 16 features and we added 4 “noise” features that were randomly generated without the decoding function. We added noise features in order to simulate real world settings that included noise. We rewrote this part of the Appendix to make it clearer. In addition, we uploaded the script for generating the dataset to the public repository and mentioned this in the manuscript.

 

Journal requirements Comments :

(Response below after all comments)

The comments: 

2. Thank you for stating the following in the Acknowledgments Section of your manuscript: [This study was supported by the Chief Scientist Office, Israeli Ministry of Health 697 (CSO-MOH, IL) as part of grant #3-000015730 within Era-PerMed. DB, CA, RF and 698 JM are shareholders or employees of Aifred Health. This work was supported by an 699Era-PerMed 2020 Grant.]

 [AK and AR were supported by the Chief Scientist Office, Israeli Ministry of Health (CSO-MOH, IL url: https://www.health.gov.il/English/Pages/HomePage.aspx) as part of grant #3-000015730 within Era-PerMed. The funders had no role in study design, data collection and analysis, decision to publish, or preparation of the manuscript.

AK and AR have received honoraria from Aifred Health (https://www.aifredhealth.com/).]

Additionally, because some of your funding information pertains to commercial funding, we ask you to provide an updated Competing Interests statement, declaring all sources of commercial funding. 

In your Competing Interests statement, please confirm that your commercial funding does not alter your adherence to PLOS ONE Editorial policies and criteria by including the following statement: "This does not alter our adherence to PLOS ONE policies on sharing data and materials.” as detailed online in our guide for authors http://journals.plos.org/plosone/s/competing-interests. If this statement is not true and your adherence to PLOS policies on sharing data and materials is altered, please explain how. Please include the updated Competing Interests Statement and Funding Statement in your cover letter. We will change the online submission form on your behalf.

3. Thank you for providing the following Funding Statement: 

[AK and AR were supported by the Chief Scientist Office, Israeli Ministry of Health (CSO-MOH, IL url: https://www.health.gov.il/English/Pages/HomePage.aspx) as part of grant #3-000015730 within Era-PerMed.The funders had no role in study design, data collection and analysis, decision to publish, or preparation of the manuscript.

AK and AR have received honoraria from Aifred Health (https://www.aifredhealth.com/). ]. 

We note that one or more of the authors is affiliated with the funding organization, indicating the funder may have had some role in the design, data collection, analysis or preparation of your manuscript for publication; in other words, the funder played an indirect role through the participation of the co-authors. 

If the funding organization did not play a role in the study design, data collection and analysis, decision to publish, or preparation of the manuscript and only provided financial support in the form of authors' salaries and/or research materials, please review your statements relating to the author contributions, and ensure you have specifically and accurately indicated the roles that these authors had in your study in the Author Contributions section of the online submission form. Please make any necessary amendments directly within this section of the online submission form. Please also update your Funding Statement to include the following statement: “The funder provided support in the form of salaries for authors [insert relevant initials], but did not have any additional role in the study design, data collection and analysis, decision to publish, or preparation of the manuscript. The specific roles of these authors are articulated in the ‘author contributions’ section.

If the funding organization did have an additional role, please state and explain that role within your Funding Statement. 

Please also provide an updated Competing Interests Statement declaring this commercial affiliation along with any other relevant declarations relating to employment, consultancy, patents, products in development, or marketed products, etc. 

Response: This research was funded by a multinational, investigator initiated grant which funded both the researchers at Bar Ilan University and through a subcontract from the Douglas University Mental Health Institute, the researchers at Aifred Health. As such in this case Aifred Health is not the primary funder of this work and is instead an industry partner. Aifred Health has provided honoraria to the Bar Ilan researchers in connection to their support of Aifred Health during the XPRIZE competition, however this was not sufficient to fund the present work. The funding agency connected with the grant had no part in the design of this project. The work itself was a full collaboration between the authors at Aifred Health and the authors at Bar Ilan, and as such Aifred Health as an industry partner was involved in every aspect of this work, though not as the primary funder. However this does not alter adherence to PLOS One standards: data is not provided as it does not belong to any of the authors; information on how to obtain it has been provided below. The model architecture is detailed enough to allow replication, and the code for running experiments is available in a public repository (we apologize- that it was not public before was an error, but it was our intention when we provided the link).

We apologize for the confusion and as such we have expanded and updated the statement as follows: 

“AK and AR were supported by the Chief Scientist Office, Israeli Ministry of Health (CSO-MOH, IL url: https://www.health.gov.il/English/Pages/HomePage.aspx) as part of grant #3-000015730 within Era-PerMed.DB, CA, JM, RF and GT were also funded by the Canadian arm of this grant (ERA-Permed Vision 2020 supporting IMADAPT), with DB, CA, JM, RF funded via their involvement in Aifred Health, which was subcontracted to complete work as part of this grant. This was the grant which served as the primary funder of this work. The funders (the granting agencies) had no role in study design, data collection and analysis, decision to publish, or preparation of the manuscript.

AK, AK and AR have received honoraria from Aifred Health (https://www.aifredhealth.com/). Aifred Health was not the primary funder of this study, and honoraria were provided in connection to the support of the Aifred Health team during the IBM Watson AI XPRIZE competition. Aifred Health-affiliated co-authors collaborated with other co-authors in the conduct of this work. JK is a member of Aifred Health’s scientific advisory board and has received stock options from Aifred Health. DB, CA, RF, JM are shareholders, employees and/or officers of Aifred Health. 

AK, AK, AR, CA, JM, RF, and DB are co-inventors on a patent pending relating to this work.”

This does not alter our adherence to PLOS ONE policies on sharing data and materials. Data is not owned by the authors, but information on how to request it is available via the data sharing statement. Sufficient information is available in the manuscript and in the associated appendices and code to reproduce the experiments described herein.

---

## [Decision Letter · Decision Letter 1]

27 Sep 2021

Treatment selection using prototyping in latent-space with application to depression treatment

PONE-D-21-18970R1

Dear Dr. Kleinerman,

We’re pleased to inform you that your manuscript has been judged scientifically suitable for publication and will be formally accepted for publication once it meets all outstanding technical requirements.

Kind regards,

Luca Citi, PhD

Academic Editor

PLOS ONE

Reviewers' comments:

Reviewer's Responses to Questions

**Comments to the Author**

1. If the authors have adequately addressed your comments raised in a previous round of review and you feel that this manuscript is now acceptable for publication, you may indicate that here to bypass the “Comments to the Author” section, enter your conflict of interest statement in the “Confidential to Editor” section, and submit your "Accept" recommendation.

Reviewer #1: (No Response)

Reviewer #2: All comments have been addressed

Reviewer #3: (No Response)

Reviewer #4: All comments have been addressed

2. Is the manuscript technically sound, and do the data support the conclusions?

Reviewer #1: Yes

Reviewer #2: (No Response)

Reviewer #3: (No Response)

Reviewer #4: Yes

3. Has the statistical analysis been performed appropriately and rigorously? 

Reviewer #1: Yes

Reviewer #2: (No Response)

Reviewer #3: (No Response)

Reviewer #4: Yes

4. Have the authors made all data underlying the findings in their manuscript fully available?

Reviewer #1: Yes

Reviewer #2: (No Response)

Reviewer #3: (No Response)

Reviewer #4: No

5. Is the manuscript presented in an intelligible fashion and written in standard English?

Reviewer #1: Yes

Reviewer #2: (No Response)

Reviewer #3: (No Response)

Reviewer #4: Yes

6. Review Comments to the Author

Reviewer #1: The response by the authors to reviewer comments on the determination of an optimal for k is 6 but in the body of paper in section 5.3.2 the value written is 5 for k instead of 6.

The full meaning of KMNN is not addressed on page 6.

Reviewer #2: (No Response)

Reviewer #3: Authors have improved the manuscript and have addressed the previous concerns

Reviewer #4: (No Response)

7. PLOS authors have the option to publish the peer review history of their article (what does this mean?). If published, this will include your full peer review and any attached files.

Reviewer #1: **Yes: **Stephen Gbenga Fashoto

Reviewer #2: No

Reviewer #3: No

Reviewer #4: No

---

## [Editor Report · Acceptance letter]

27 Oct 2021

PONE-D-21-18970R1 

Treatment selection using prototyping in latent-space with application to depression treatment 

Dear Dr. Kleinerman:

I'm pleased to inform you that your manuscript has been deemed suitable for publication in PLOS ONE. Congratulations! Your manuscript is now with our production department. 

Kind regards, 

on behalf of

Dr. Luca Citi 

Academic Editor

PLOS ONE